# CardioComposer: Leveraging Differentiable Geometry for Compositional Control of Anatomical Diffusion Models

**Karim Kadry**[*]   Shoaib Goraya[†]   Ajay Manicka[*]   Abdalla Abdelwahed[‡]
Naravich Chutisilp [§]   Farhad R. Nezami[†]   Elazer R. Edelman[*]

## Abstract

Generative models of 3D cardiovascular anatomy can synthesize informative structures for clinical research and medical device evaluation, but face a trade-off between geometric controllability and realism. We propose CardioComposer: a programmable, inference-time framework for generating multi-class anatomical label maps from interpretable ellipsoidal primitives. These primitives represent geometric attributes such as the size, shape, and position of discrete substructures. We specifically develop differentiable measurement functions based on voxel-wise geometric moments, enabling loss-based gradient guidance during diffusion model sampling. We demonstrate that these losses can constrain individual geometric attributes in a disentangled manner and provide compositional control over multiple substructures. Finally, we show that our method is compatible with a broad range of anatomical systems containing non-convex substructures, spanning cardiac, vascular, and skeletal organs. We release our code at https://github.com/kkadry/CardioComposer.

## 1 Introduction

Three-dimensional segmentations of human anatomy power a variety of physics-based simulation platforms. For example, virtual cohorts of anatomy can be used for virtual clinical trials to evaluate and optimize novel medical devices and imaging systems (Sarrami-Foroushani et al., 2021; Viceconti et al., 2021; Abadi et al., 2020). Additionally, biophysical simulations can generate insights in the context of both computational physiology research (Niederer et al., 2020; Straughan et al., 2023; Roney et al., 2020) and surgical training (Yu et al., 2024). Anatomical segmentations can also be used to augment machine-learning workflows through the formation of synthetic images, either through imaging simulators (Gopalakrishnan et al., 2024; Gopalakrishnan & Golland, 2022), domain randomization (Dey et al., 2025; Billot et al., 2023), or generative models (Fernandez et al., 2022; 2024).

Generative models of anatomy trained on patient-specific data offer advantages for simulation use-cases. For example, conditional generation can augment computational trial cohorts with anatomical variants that are both novel and rare (Kong et al., 2024). Moreover, generative editing methods, such as inpainting, can precisely modify existing patient geometries to create anatomically plausible variations (Kadry et al., 2024; 2025). These "digital siblings" can be used with biophysical simulators to model *counterfactual* scenarios that elucidate the relationship between anatomical form and function.

However, unlike generative modeling of 3D shapes for artistic purposes, generating anatomical models for biophysical simulations presents several unique challenges. The first concerns *scale-critical* features, in which minor geometric variations on the order of millimeters can cause major fluctuations in physiological behavior (Fabris et al., 2022; Sacco et al., 2018; Moore & Dasi, 2015).

[*]Massachusetts Institute of Technology
[†]Brigham and Women's Hospital
[‡]American University in Cairo
[§]École Polytechnique Fédérale de Lausanne

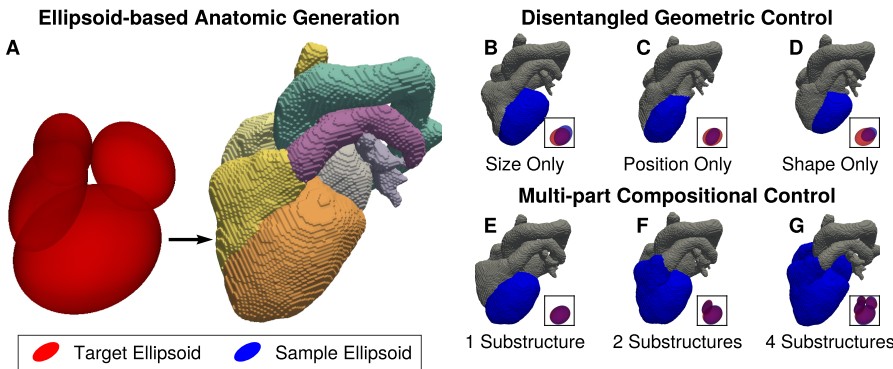

Figure 1: **We present a guidance framework to constrain diffusion models of multi-label anatomical segmentations based on simple geometric features**. Such features include size, shape, and position, and can be represented as ellipsoids in 3D space (panel A). Our inference-time approach enables generation based on independent geometric features (panels B-D), and supports multi-component compositional generation (panels E-G). Gray ⬜ and blue 🟦 voxels represent components that are unconstrained and constrained, respectively. Purple ellipsoids 🟣 indicate a strong overlap between target 🔴 and sample ellipsoids 🔵 .

Second, anatomical structure exerts *attribute-specific* effects, in which geometric features such as size and position play different roles in determining biophysical outcomes (Kadry et al., 2021). Third, the geometry of multiple substructures interact in a *compositional* manner (Kadry et al., 2021; Bhalodia et al., 2018; Kong et al., 2024), where simulated outcomes are influenced by the collective arrangement of various substructures. Lastly, to interface with clinicians and device engineers, such generative models should be controllable via primitives that are interpretable and physiologically relevant.

To address these design requirements, we present CardioComposer, an energy-based guidance framework for controlling unconditional diffusion models with geometric attributes regarding size, shape, and position. We visually represent these constraints via interpretable ellipsoidal primitives (Figure 1 A). Our inference-time framework can independently control individual attributes such as size or position (Figure 1 B-D), and compose geometric constraints for an arbitrary number of anatomical components or substructures (Figure 1 E-G). Our *key insight* is that unconditional diffusion models of multi-class anatomy can be constrained in a compositional manner by simple gradients derived from geometric loss functions applied individually to each substructure. We demonstrate this method on multi-tissue cardiovascular segmentations that exhibit a wide array of substructures such as star-shaped chambers and tubular vasculature. Our framework advances the state of the art in the following ways:

- **Differentiable Geometry for Anatomical Characterization**: We introduce a set of differentiable geometric measurement functions that compute physiologically relevant anatomical features from a substructure label map. We specifically measure voxel-wise geometric moments, computing size via zeroth-order moments, position via first-order moments, and shape via scale-normalized second-order moments.

- **Inference-time Guidance to Control Substructure Geometry**: We demonstrate that simple gradients derived from differentiable geometric loss functions can guide unconditional latent diffusion models of discretized multi-class label maps. This enables *independent* or *joint* control of substructure attributes without retraining, where substructures consist of one tissue class or the union of multiple classes.

- **Complex Compositional Control**: We validate that multiple substructure-specific geometric losses can be composed to enable more complex anatomical constraints. Further, we show that this control extends to non-convex substructures with branching or curved geometry.

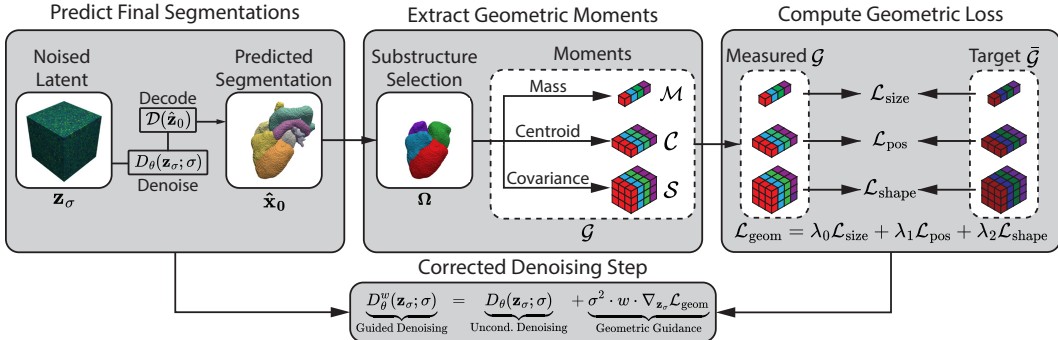

Figure 2: **Our method involves applying a geometric guidance correction step for every denoising iteration**. Left: The noised latent $\mathbf{z}_\sigma$ is passed through the diffusion model and VAE decoder to produce a clean voxel space prediction $\hat{\mathbf{x}}_0$ (Section 4.2). Middle: The segmentation is parsed for relevant substructures $\Omega$, and geometric moments $\mathcal{G}$ are extracted for each substructure (Section 4.3). Right: Measured geometric moments $\mathcal{G}$ are compared to target moments $\bar{\mathcal{G}}$ through geometric moment losses. Bottom: The gradient derived from the aggregate loss corrects the denoising step.

## 2  BACKGROUND

**Traditional Morphometric Modeling for Anatomy.** Morphometry involves quantifying anatomical structure through geometric measurements. Traditional morphometric approaches measure intrinsic features such as length, area, volume, and shape, as well as extrinsic factors such as position and orientation. Geometric measurements enable applications such as cardiovascular risk stratification (Asheghan et al., 2023; Mahmod et al., 2024) and orthopedic diagnosis (Gatti et al., 2024). However, traditional morphometric approaches face two key challenges: they cannot represent complex relationships between features, and multiple distinct anatomies may map to the same high-level measurements. To address these limitations, we propose a framework in which an unconditional diffusion model is controlled by traditional morphometric features (size, shape, and position) to generate a variety of realistic anatomical structures, providing an approachable anatomical modeling interface for both clinical and engineering workflows involving numerical simulation.

**Generative Models for Numerical Simulation.** Anatomical models in the form of 3D meshes or label maps serve as a crucial tool for studying form-function relationships through physical simulations, enabling both scientific discovery and medical device design. However, current approaches to create such models must trade off between *fidelity* and *control*. Simple geometric primitives, such as cylinders for coronary arteries (Dong et al., 2023) and truncated ellipsoids for cardiac chambers (Aróstica et al., 2025) offer parametric control but fail to capture anatomical realism. Data-driven approaches such as autoencoders (Dou et al., 2024; Qiao et al., 2025) represent anatomy in terms of global shape vectors, and can generate synthetic data for mechanistic studies of heart disease (Hermida et al., 2024; Williams et al., 2022). However, such approaches are limited in their ability to model *interpretable* geometric attributes. Deformation editing methods (Pham et al., 2023; 2024) allow for interpretable control of anatomical geometry, but are limited to modifying existing structures. Recently, diffusion-based approaches such as inpainting and partial diffusion have been used to edit patient-specific anatomy to create "digital siblings" (Kadry et al., 2024). However, such edits can induce undesirable morphological bias when applied to rare and pathological cases. To this end, recent studies have imposed anatomical features by explicitly providing scalar conditioning features during training. For example, de Wilde et al. (2025) trained a conditional model on thyroid segmentations, and Kadry et al. (2025) introduced morpho-skeletal conditioning and guidance mechanisms for coronary arteries. However, both approaches rely on conditional training and are restricted to size-related variables such as volume or cross-sectional area. Similarly, Du et al. (2025) presented a hierarchical conditional diffusion model for generating aortic centerlines and radial profiles, but is restricted to fixed centerline connectivity and cannot flexibly accommodate topological changes such as varying branching patterns. In contrast, we propose a modular inference-time framework that controls *unconditional* diffusion models across diverse anatomical structures using geometric attributes such as size, shape, and position.

**Spatial Control of Generative Models.** Spatial control of generative models is achieved through two principal approaches. The first approach conditions models on interpretable mid-level representations (e.g., bounding boxes, ellipsoid parameters, articulation angles) and has been successfully applied to images (Nie et al., 2024), videos (Feng et al., 2025), 3D objects (Hertz et al., 2022; Koo et al., 2023; Mu et al., 2021), and proteins (Stark et al., 2025). However, these methods cannot accommodate novel constraints without retraining. The second approach employs energy-based guidance during the reverse diffusion process (Bansal et al., 2023; Du et al., 2023), enabling flexible constraint composition at test time, but is typically limited to general localization rather than exact geometric control. Recent works such as self-guidance use attention-based loss functions to enable basic geometric attribute control (size, position) in text-to-image models (Epstein et al., 2023), but are not designed for multi-label segmentations, nor do they control for orientation or aspect ratio. In our work, we extend energy-based guidance by introducing differentiable geometric losses for 3D multi-component anatomical voxel maps based on substructure-specific geometric properties such as the mass, centroid, and covariance, enabling the composition of multiple constraints across several independent substructures.

## 3 ANATOMICAL DIFFUSION MODELS

Let $\mathbf{x} \in \mathbb{R}^{C \times H \times W \times D}$ be a 3D segmentation volume with $C$ tissue channels and $(H, W, D)$ spatial dimensions. We employ a variational autoencoder (VAE) with an encoder $\mathcal{E}$ that maps $\mathbf{x}$ to a lower-dimensional latent representation $\mathbf{z} = \mathcal{E}(\mathbf{x})$, and a decoder $\mathcal{D}$ that maps $\mathbf{z}$ back to a voxel-space reconstruction $\tilde{\mathbf{x}} = \mathcal{D}(\mathbf{z})$. The latent grid $\mathbf{z} \in \mathbb{R}^{c \times h \times w \times d}$ comprises $c$ channels and spatial dimensions $(h, w, d) = (H/f, W/f, D/f)$ for an integer downsampling factor $f$.

We use an unconditional latent diffusion model (LDM) as a prior over 3D anatomical segmentations, trained on the encoded latent representations $\mathbf{z}$. We specifically use the elucidated diffusion formulation of Karras et al. (2022). In the forward process, data samples $\mathbf{z} \sim p_{\text{data}}(\mathbf{z})$ are progressively corrupted by adding Gaussian noise, resulting in perturbed data $\mathbf{z}_\sigma = \mathbf{z} + \mathbf{n}$ where $\mathbf{n} \sim \mathcal{N}(\mathbf{0}, \sigma^2 \mathbf{I})$. The reverse process reconstructs the original data by approximating the score function $\nabla_{\mathbf{z}_\sigma} \log p(\mathbf{z}_\sigma; \sigma)$ that controls the reverse diffusion process:

$$d\mathbf{z}_\sigma = -2\sigma \nabla_{\mathbf{z}_\sigma} \log p(\mathbf{z}_\sigma; \sigma)\, dt + \sqrt{2\sigma}\, d\mathbf{w} \tag{1}$$

where $d\mathbf{w}$ is the Wiener process. This score function $\nabla_{\mathbf{z}_\sigma} \log p(\mathbf{z}_\sigma; \sigma) = (D_\theta(\mathbf{z}_\sigma; \sigma) - \mathbf{z}_\sigma)/\sigma^2$ can be expressed via a denoising function $D_\theta$ parametrized by a 3D U-Net $F_\theta$ through the following relation:

$$D_\theta(\mathbf{z}_\sigma; \sigma) = c_{\text{skip}}(\sigma)\, \mathbf{z}_\sigma + c_{\text{out}}(\sigma)\, F_\theta(c_{\text{in}}(\sigma)\, \mathbf{z}_\sigma; c_{\text{noise}}(\sigma)), \tag{2}$$

where $(c_{\text{skip}}, c_{\text{out}}, c_{\text{in}}, c_{\text{noise}})$ are noise-level–dependent scaling coefficients. The neural network is trained by minimizing the clean-data prediction objective $L = \mathbb{E}_{\sigma, \mathbf{z}, \mathbf{n}} \left[ \lambda(\sigma) \| D_\theta(\mathbf{z}_\sigma; \sigma) - \mathbf{z} \|_2^2 \right]$, with $\lambda(\sigma)$ balancing loss contributions across noise levels.

## 4 GEOMETRIC GUIDANCE

### 4.1 OVERVIEW

Our objective is to guide an unconditional diffusion model that synthesizes anatomical segmentations with geometric constraints for size, position, and shape. These attributes are measured on substructures that correspond to discrete tissue labels within the 3D voxel map. To do this, we guide the sampling process with a composite *geometric loss* applied to a subset of labels. This geometric loss is a weighted sum of moment-based terms: size via zeroth-order moments (scalar volumes), position via first-order moments (centroid vectors), and shape via scale-invariant second-order moments (normalized covariance matrices). Figure 2 illustrates four main stages. First, at each sampling step we denoise the latent, decode to voxel-space logits, and apply a softmax to obtain class probabilities. Second, we select the desired anatomical substructures $\mathbf{\Omega}$ and extract the geometric moments $\mathcal{G} = [\mathcal{M}, \mathcal{C}, \mathcal{S}]$, representing the mass, centroid, and covariance for each substructure. Third, we compute the geometric loss $\mathcal{L}_{\text{geom}}$ with respect to target moments $\bar{\mathcal{G}}$. Lastly, the gradient of this loss with respect to the noisy latents is used to guide the sampling process.

## 4.2 Segmentation Denoising and Guidance

We formulate loss-based guidance in terms analogous to diffusion posterior sampling (Chung et al., 2023), where the gradient derived from a differentiable geometric loss $\mathcal{L}_{\text{geom}}$ guides the sampling process. To guide anatomical generation, the intermediately noised latent $\mathbf{z}_\sigma$ is denoised by the diffusion model to produce a clean prediction $\hat{\mathbf{z}}_0 = D_\theta(\mathbf{z}_\sigma; \sigma)$ and subsequently decoded into a voxel-space segmentation $\hat{\mathbf{x}}_0 = \mathcal{D}(\hat{\mathbf{z}}_0)$. As the decoder outputs are continuous logits, we apply a label-wise softmax to ensure that the segmentation values are between 0 and 1. The geometric loss $\mathcal{L}_{\text{geom}}$ is then computed in a differentiable manner to update the denoiser predictions through the gradient with respect to the noised latent $\mathbf{z}_\sigma$. The update step is parameterized with a guidance weight $w$ as follows:

$$\underbrace{D_\theta^w(\mathbf{z}_\sigma; \sigma)}_{\text{Guided Denoising}} = \underbrace{D_\theta(\mathbf{z}_\sigma; \sigma)}_{\text{Uncond. Denoising}} - \underbrace{\sigma^2 \cdot w \cdot \nabla_{\mathbf{z}_\sigma} \mathcal{L}_{\text{geom}}}_{\text{Geometric Guidance}} \tag{3}$$

## 4.3 Geometric Moment Loss

To isolate guidance to specific substructures representing individual tissues, we map the input segmentation $\hat{\mathbf{x}}_0 \in \mathbb{R}^{C \times H \times W \times D}$ to a set of substructure voxel maps $\boldsymbol{\Omega} \in \mathbb{R}^{E \times H \times W \times D}$. Here, $E$ specifies the number of relevant substructures. Substructures are determined either through taking subsets of the tissue channels or taking the union of multiple tissue channels.

To extract geometric features, we compute the set of geometric moments $\mathcal{G} = [\mathcal{M}, \mathcal{C}, \mathcal{S}]$, where $\mathcal{M} \in \mathbb{R}^{E \times 1}$ represents the masses or volumes for each substructure, $\mathcal{C} \in \mathbb{R}^{E \times 3}$ represents the centroids, and $\mathcal{S} \in \mathbb{R}^{E \times 3 \times 3}$ represents the covariances. Specifically, for each individual substructure index $k$, we define $\Omega_k \in \mathbb{R}^{(H \times W \times D) \times 1}$ as the flattened substructure voxel grid and $\mathbf{p} \in \mathbb{R}^{(H \times W \times D) \times 3}$ as the normalized voxel coordinates between 0 and 1. We compute the geometric moments as

$$\mathcal{M}_k = \mathbf{1}^T \cdot \Omega_k \quad \text{and} \quad \mathcal{C}_k = \frac{\Omega_k^T \mathbf{p}}{\mathcal{M}_k} \quad \text{and} \quad \mathcal{S}_k = \frac{1}{\mathcal{M}_k} \mathbf{p}^T \text{diag}(\Omega_k) \, \mathbf{p} - \mathcal{C}_k^T \mathcal{C}_k \tag{4}$$

where $\mathbf{1}^T$ is the all-ones vector, and $diag(\cdot)$ refers to a diagonal matrix embedding. To enable independent control over size and shape characteristics, we compute a normalized representation of the covariance matrix. The scale-normalized covariance matrix is defined as $\mathcal{S}_k^n = \mathcal{S}_k / \text{tr}(\Lambda)$ where $\Lambda$ is the eigenvalue matrix obtained from the eigendecomposition of $\mathcal{S}_k$. Intuitively, the normalized covariance matrix represents the aspect ratio and orientation of the substructure.

Following the computation of geometric moments, we calculate individual loss terms by comparing each moment to its corresponding target moment $\bar{\mathcal{G}} = [\bar{\mathcal{M}}, \bar{\mathcal{C}}, \bar{\mathcal{S}}^n]$. For each geometric feature, we compute the mean squared error (MSE) between the measured and target values. These individual loss terms are defined as:

$$\mathcal{L}_{\text{size}} = \mathcal{L}_{\text{MSE}}(\mathcal{M}, \bar{\mathcal{M}}), \quad \mathcal{L}_{\text{pos}} = \mathcal{L}_{\text{MSE}}(\mathcal{C}, \bar{\mathcal{C}}), \quad \mathcal{L}_{\text{shape}} = \mathcal{L}_{\text{MSE}}(\mathcal{S}^n, \bar{\mathcal{S}}^n). \tag{5}$$

Using prescribed weight factors $\lambda_0, \lambda_1, \lambda_2$, we compute the aggregate geometric loss as $\mathcal{L}_{\text{geom}} = \lambda_0 \mathcal{L}_{\text{size}} + \lambda_1 \mathcal{L}_{\text{pos}} + \lambda_2 \mathcal{L}_{\text{shape}}$. The weighted sum of each weight $\lambda_i$ allows us to control the contribution of each individual loss to the guidance process, enabling easy disentangled control by zeroing out the associated weighting factor.

## 5 Experiments

### 5.1 Unconditional Model Training

For diffusion training, we use the label maps provided in the TotalSegmentator dataset (Wasserthal et al., 2023). We extract heart-related labels, which include aorta (Ao), pulmonary artery (PA), pulmonary veins (PV), inferior vena cava (IVC), superior vena cava (SVC), left atrium (LA), right atrium (RA), left ventricle (LV), right ventricle (RV), and left ventricular myocardium (Myo). We manually filter out low-quality label maps, resulting in 596 3D cardiac segmentations with 11 channels and an isotropic voxel edge length of $2\,\text{mm}$ (Section 8.2). We split the dataset into training and validation sets with an 80/20 split. All target moments and evaluation metrics are computed on the validation set.

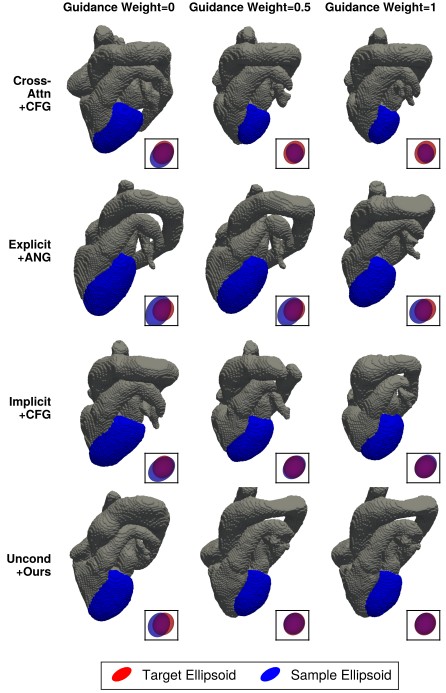

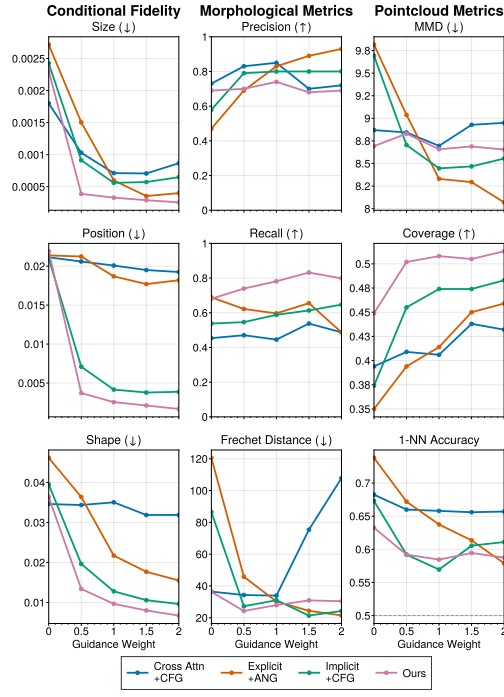

Figure 3: **Geometric guidance can generate synthetic anatomy with geometric constraints.** Grid shows example synthetic label maps where constraints are applied to the myocardium voxels ■. Rows: baseline conditioning and guidance methods (CFG = classifier-free guidance, ANG = adaptive null guidance).

Figure 4: **Geometric guidance can enforce conditional fidelity while maintaining realism.** Line plots compare conditioning and guidance mechanisms based on the geometric properties of the myocardium. MMD values are multiplied by $10^3$.

We train an unconditional diffusion model on cardiac label maps similarly to Kadry et al. (2024) (further details in Section 8.3). To compute our geometric guidance loss, we use the weighted sum of the individual geometric moment losses, where the guidance weights $\lambda_i$ are tuned experimentally (Section 8.4).

## 5.2 BASELINES

We compare our approach (unconditional diffusion combined with geometric guidance) to conditional training approaches. Given the target geometric moments representing the size $\mathcal{M}$, centroid $\mathcal{C}$, and covariance $\mathcal{S}$ of each cardiac substructure, we condition the model in the following ways:

- **Explicit Concatenation**: We directly encode geometric attributes as scalar values in the conditioning signal (Kadry et al., 2025). Here, we adapt this method to positional and shape-based features. We flatten and stack all geometric moments into a 13-dimensional vector for all $E$ substructures. We then expand this vector into a voxel grid $\mathcal{G}_{\mathrm{exp}} \in \mathbb{R}^{(13 \times E) \times h \times w \times d}$ which is concatenated to the latents along the channel dimension.

- **Implicit Concatenation**: We indirectly encode geometric attributes in the conditioning signal through 3D heatmaps (Kadry et al., 2025). Here, we embed geometric moments as 3D Gaussians in voxel space. For each substructure, we create a voxel map $\mathcal{G}_{\mathrm{imp}} \in \mathbb{R}^{E \times h \times w \times d}$ where the voxel values encode the Mahalanobis distance.

- **Cross-attention**: We express the conditioning signal as a sequence of tokens where each token represents substructure geometry. The dimension of each token corresponds to the embedded geometric moments $\mathcal{G}_{\mathrm{cross}} \in \mathbb{R}^{E \times 256}$. To enable sequence conditioning for the denoising U-Net, we convert the self-attention layers to cross-attention layers, similar to Rombach et al. (2022).

We implement guidance mechanisms such as adaptive null guidance (ANG) (Kadry et al., 2025) for explicit concatenation, and classifier-free guidance (CFG) (Ho & Salimans, 2022) for implicit concatenation and cross-attention. Further details can be found in Section 8.5.

## 5.3 EVALUATION METRICS

We evaluate pairwise conditional fidelity for size, shape, and position by taking the $L_1$-norm between the target and sample moments, averaging over all relevant substructures. We measure morphological quality metrics by comparing the distribution of real and synthetic anatomy in morphological feature space (Kadry et al., 2024). To embed each label map, we consider all 10 tissues as substructures and concatenate, over all substructures, the masses, centroids, and eigenvalues of the normalized covariance matrices. We specifically use morphological variants of improved precision and recall, as well as the Fréchet distance (FD) (Kynkäänniemi et al., 2019; Kadry et al., 2024). Lastly, we leverage pointcloud-based metrics to assess 3D shape (Yang et al., 2019), such as minimum matching distance (MMD), coverage (COV), and 1-nearest neighbor accuracy (1-NNA). Distances between pointclouds are computed with Earth Mover's Distance (EMD). Further details can be found in Section 8.6.

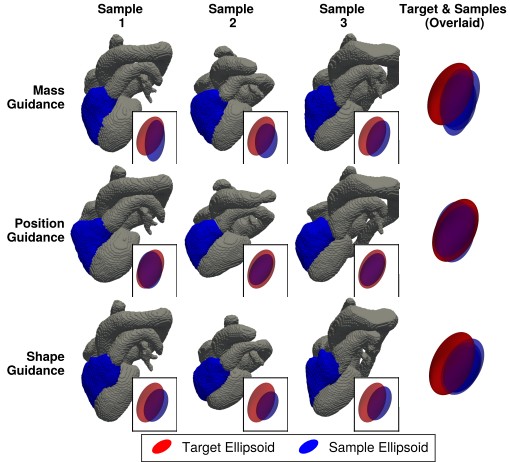

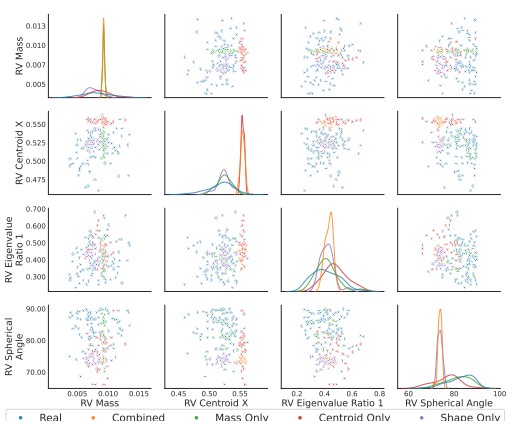

Figure 5: **Geometric guidance enables independent control of size, shape, and position.** Columns show synthetic label maps generated by geometric guidance applied to the right ventricle voxels ■ using various geometric losses. Rows represent which geometric feature is being independently controlled.

Figure 6: **Geometric guidance enables independent control of size, shape, and position.** Pair plot shows kernel density estimate plots (diagonals) and pairwise scatterplots (off-diagonals) of morphological metrics. Guidance is applied to control the right ventricle geometry.

## 5.4 EVALUATING ANATOMICAL GENERATION QUALITY

We first aim to compare and evaluate geometric control methods on both conditional fidelity and synthetic anatomy quality. We sample target moments for a single substructure (myocardium) from the validation set and generate 200 anatomical segmentations per method. We sweep over guidance weights $w \in [0, 2]$. In Figure 4, we show that geometric guidance enhances conditional fidelity, especially at higher guidance weights. We observe that our method maintains generation quality, retaining

Table 1: Comparative analysis of various approaches for multi-substructures compositional generation. The number of substructures indicates the number of tissues actively constrained during sampling. MMD values are multiplied by $10^3$.

| Constraints | Method | Morph. Metrics | | | Pointcloud Metrics | | |
|---|---|---|---|---|---|---|---|
| | | FD ($\downarrow$) | Pr. ($\uparrow$) | Re. ($\uparrow$) | MMD ($\downarrow$) | COV ($\uparrow$) | 1-NNA |
| 0 | Implicit | 1622 | 0.00 | **0.99** | 55.7 | 0.288 | 0.915 |
| | Ours | **34.6** | **0.70** | 0.87 | **9.40** | **0.53** | **0.55** |
| 1 | Implicit | 227 | 0.00 | **0.87** | 17.1 | 0.40 | 0.79 |
| | Ours | **38.5** | **0.60** | 0.83 | **9.39** | **0.52** | **0.57** |
| 3 | Implicit | **29.8** | **0.80** | 0.81 | 9.21 | 0.48 | 0.58 |
| | Ours | 32.7 | 0.78 | **0.94** | **8.60** | **0.58** | **0.52** |
| 6 | Implicit | **31.1** | **0.82** | **0.95** | **8.11** | 0.56 | 0.50 |
| | Ours | 35.5 | 0.80 | 0.94 | 8.50 | **0.58** | 0.50 |

similar levels of morphological and pointcloud evaluation metrics with increasing guidance. Figure 3 shows example label maps generated through varying guidance values for all methods; only our method and implicit conditioning align the target and sample ellipsoids under guidance. Further information on which features were plotted can be found in Section 8.13.

## 5.5 Evaluating Geometric Disentanglement

We next show that our guidance framework uniquely enables disentangled control of geometric attributes. We use 100 target moments for myocardial labels from the validation set using no losses (Uncond.), a combination of all losses ($\mathcal{L}_{\text{geom}}$), or each individual moment loss ($\mathcal{L}_{\text{size}}$, $\mathcal{L}_{\text{pos}}$, and $\mathcal{L}_{\text{shape}}$). Figure 9 shows that each individual loss improves its corresponding conditional-fidelity metric while leaving the others approximately unchanged. The main exception is the interaction between shape and mass, where adding a guidance weight for the shape loss enhances mass fidelity. This phenomenon is likely due to correlations between size and shape in the dataset. Qualitative results can be seen in Figures 5 and 6, where the right ventricle is constrained independently by mass, position, or shape. For example, mass-only guidance produces a narrow peak in the mass marginal while the other morphology metrics remain broad, whereas applying all geometric losses collapses all marginals to narrow peaks at their target values.

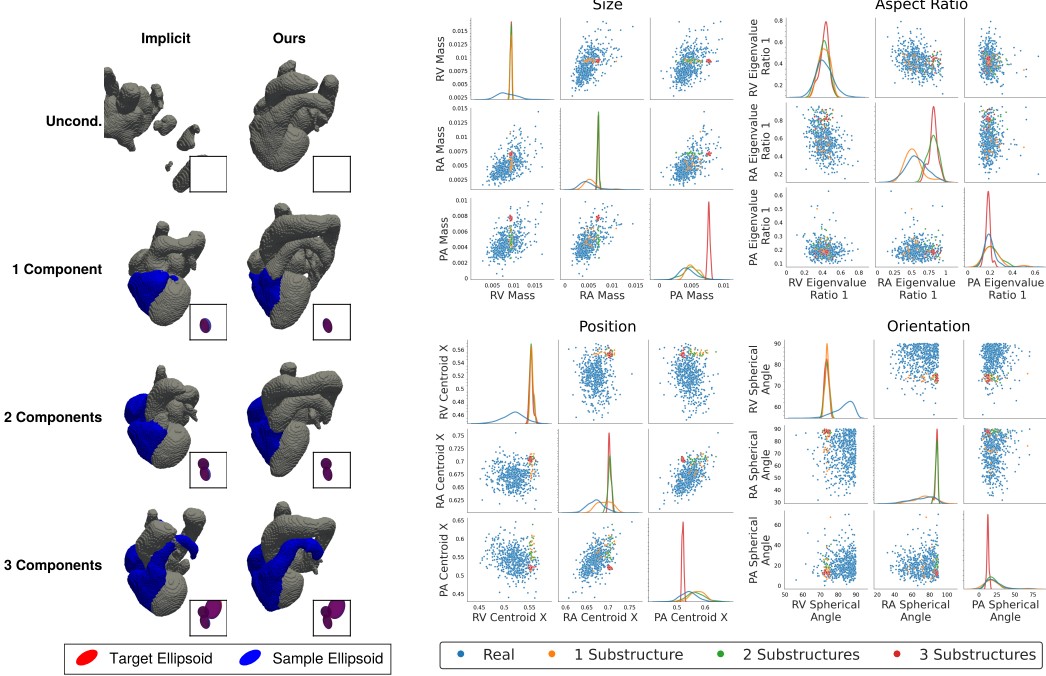

Figure 7: **Geometric guidance exhibits enhanced multi-part compositional generation compared to a conditional drop-out baseline.** Columns: Baseline vs. our method. Rows: Synthetic label maps with a varying number of voxel labels ■.

Figure 8: **Our guidance framework enables multi-part compositional generation.** Pair plot shows kernel density estimate plots (diagonals) and pairwise scatterplots (off-diagonals) for various morphological metrics. Guidance is applied to control the geometry for a varying number of substructures.

## 5.6 Evaluating Multi-Part Compositionality

We evaluate the ability of our method to achieve multi-part control under arbitrary constraints. We sample 100 target moment sets from the validation set and constrain generation based on: (a) only the myocardium (1 substructure), (b) the right heart labels (3 substructures), and (c) both right and left heart labels (6 substructures). For geometric guidance, we use an unconditional model and select the appropriate substructures $\Omega$ during guidance. For our baseline, we retrain the best

conditional diffusion model (implicit) with 6 substructures using dropout (further details can be found in Section 8.6). Results are shown in Table 1 and Figure 7, which show that with a small number of constrained substructures, implicit conditioning with dropout fails to generate high-quality anatomy as measured by morphological and pointcloud metrics. Because the implicit conditional baseline is trained with independent dropout over six ellipsoidal conditioning channels, the fully conditioned case (all channels present) is vastly more frequent than the unconditional case (all channels empty). As a result, unconditional sampling corresponds to the rarest training configuration and yields degraded anatomical quality in Figure 7 and Table 1.

We further show in Figure 8 that controlling multiple substructures via geometric guidance can effectively sample from lower-dimensional slices of the original morphological distribution. For instance, when guidance is applied to a single substructure, the pair plots show a sharp concentration around the target value for the right ventricle, while the remaining structures retain broad distributions. When three substructures are guided simultaneously, the corresponding morphological marginals all collapse to narrow peaks at their target geometric values. Finally, Figure 10 shows that our guidance framework applies to complex, non–star-shaped geometries, including curved and branching substructures, as well as Boolean unions involving multiple tissue classes considered as a single substructure (e.g., both vena cavae or all chambers).

## 5.7 Geometric Inpainting and Biophysical Simulations

We demonstrate that our geometric guidance framework can controllably edit patient-specific anatomy for simulation experiments. We consider an example involving biventricular pressurization in which we edit a label map to enlarge or shrink the RV. As shown in Figure 11, we define the RV target geometry by doubling or halving the mass measured from the original label map (left column insets). We apply tissue-based inpainting (Kadry et al., 2024) with geometric guidance to edit the RV (left column) and convert the label map to a tetrahedral mesh (middle column). We simulate biventricular pressurization for the baseline patient and edited variants, showing how RV volume modulates wall displacement (right column). Further details can be found in Section 8.7.

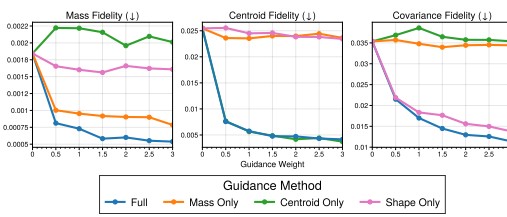

Figure 9: **Geometric guidance enables independent control of size, shape, and position.** Line plots compare conditional fidelity for individual losses (mass, position, shape) and all losses (full). Losses were computed for myocardial tissue.

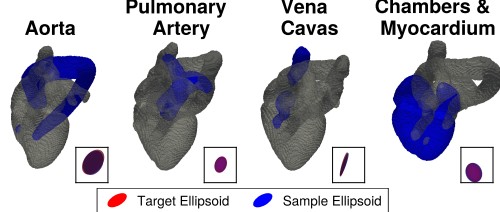

Figure 10: **Geometric guidance is compatible with complex substructures.** Qualitative results showing geometric control of substructures with non-convex or branched features, as well as substructures comprising multiple tissues.

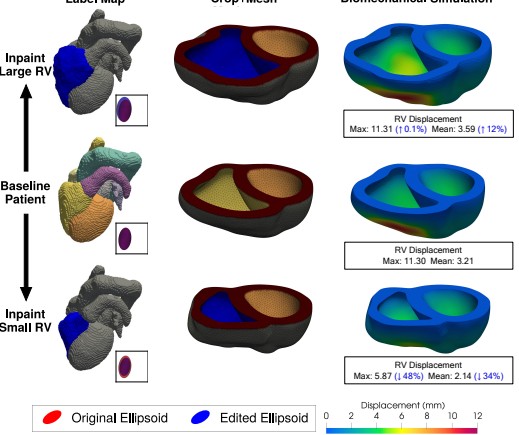

Figure 11: **Geometric guidance can controllably inpaint anatomical features for counterfactual biomechanical simulations.** Left column: a baseline patient edited to vary right-ventricle ■ size, while maintaining all other substructures ■. Middle column: cropped biventricular mesh from each scenario. Right column: editing right-ventricle size while retaining the left ventricle modulates biomechanical outcomes.

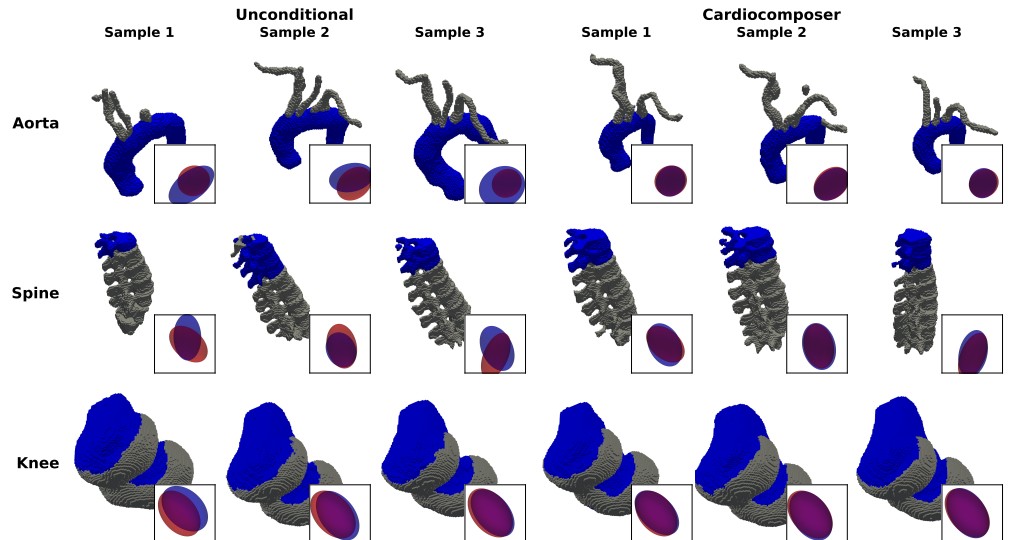

Figure 12: **Geometric guidance can control the generation of a wide variety of anatomical systems.** We present label maps that were generated from guided and unguided latent diffusion models. For the aortic dataset, we control the main trunk, for the spinal dataset, we control the sixth, seventh, and eighth thoracic vertebrae, while for the knee dataset, we control the femur.

## 5.8 GENERALITY OVER ANATOMICAL SYSTEMS AND STRUCTURES

We aim to show that geometric guidance applies to a wide variety of anatomical diffusion models. We construct three datasets of 3D multi-class patient-specific label maps corresponding to the branched ascending aorta, spinal vertebral column, and distal femur. Further details can be found in Section 8.2. In Figure 12, we show typical unconditional samples, as well as samples from geometric guidance, which constrains substructure geometry to a high degree of fidelity.

## 6 LIMITATIONS

Our method has several limitations. First, the relative weights of the geometric moments should be obtained through experimental tuning, similar to all guidance frameworks. However, we found that the same set of loss weightings transfer well to entirely different anatomical systems such as the aorta, spinal column, and knee, indicating that only minimal additional tuning is required. Second, substructures are currently defined based on label map class, and cannot represent localized features such as cross sections. Lastly, anatomical diffusion models can generate topologically incorrect substructures, such as disconnected aortas or several left atria, making the resulting simulation physics inaccurate. This can be addressed by filtering out topologically incorrect anatomies, at the cost of some wasted computation.

## 7 CONCLUSIONS

We present a flexible method to impose geometric constraints on diffusion models of 3D multi-class anatomical label maps. By measuring geometric moments relating to size, shape, and position of various substructures during inference, we enable energy-based guidance without conditional training. We show that our framework can independently control geometric attributes such as size, position, or shape, and constrain multiple anatomical substructures in a compositional manner. We also demonstrate geometric guidance across a wide range of anatomical systems and structures, spanning cardiac, vascular, and skeletal systems. Our framework enables custom-tailoring realistic anatomy for computational simulation experiments, elucidating the causal relationships between form and simulated function.

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

## 8 APPENDIX

### 8.1 OVERVIEW

- In Section 8.2, we provide details on dataset curation and processing.
- In Section 8.3, we provide implementation details for our autoencoder and diffusion model.
- In Section 8.4, we provide implementation details for our guidance algorithm.
- In Section 8.5, we provide implementation details for our conditional generation baselines.
- In Section 8.6, we provide further experimental details for evaluation and inference.
- In Section 8.7, we provide implementation details for our biomechanical simulations.
- In Section 8.8 we provide a dataset scaling analysis for our latent diffusion model.
- In Section 8.9 we provide qualitative results for a procedurally generated ellipsoid dataset.
- In Section 8.10 we provide an autoencoder reconstruction error analysis.
- In Section 8.11 we provide a scale-factor sweep analysis for target modification.
- In Section 8.12 we provide quantitative results demonstrating disentangled generation for alternative geometric features derived from the second-order moment.
- In Section 8.13, we present additional morphological distribution plots that examine the effect of guidance weight as well as the choice of control technique.

### 8.2 DATASETS

For our study, we construct four separate datasets of anatomical segmentations to qualitatively demonstrate the flexibility of geometric guidance. These datasets represent 1) whole-heart cardiac segmentations with great vessels, 2) the branched ascending aorta, 3) multi-vertebral spinal column, and 4) the femoral condyle and articular cartilage. We primarily use the cardiac dataset for our experiments.

For the cardiac dataset, we utilize TotalSegmentator v2 (Wasserthal et al., 2023), with 596 cases manually selected based on segmentation quality assessment. Cardiac structures including the myocardium (Myo), left and right atria (LA & RA), left and right ventricles (LV & RV), aorta (Ao), and pulmonary artery (PA) were segmented using a specialized TotalSegmentator model trained on sub-millimeter resolution data. For the inferior vena cava (IVC), superior vena cava (SVC), and pulmonary veins (PV), we retain the labels from the original dataset. To ensure anatomical validity, we perform topological filtration on all structures except the pulmonary veins, where filtration involves extracting only the largest connected component. The resulting segmentations are standardized by resampling to a uniform voxel resolution of 2mm and subsequently cropped to a fixed range. The crop center is determined from the union of all four chamber segmentations, and the crop size is $128^3$ voxels.

For the aorta dataset, we extract labels directly from the original TotalSegmentator v2 (Wasserthal et al., 2023) segmentations, without applying a specialized model, resulting in 450 3D segmentations manually selected based on segmentation quality assessment. The labels include the main aortic trunk and the ascending branches, which comprise the brachiocephalic trunk (BCT), left common carotid artery (LCCA), right common carotid artery (RCCA), left subclavian artery (LSCA), and right subclavian artery (RSCA), for a total of 7 channels per segmentation. All segmentations are resampled to an isotropic voxel size of $2\,\mathrm{mm}$ and cropped to a spatial size of $128^3$ using a crop center determined from the center of all combined tissues.

For the spinal dataset, we utilize the CTSpine1K dataset (Deng et al., 2021) and extract all vertebral body segmentations, resulting in 784 3D segmentations. The segmentations include 7 cervical vertebrae (C1–C7), 12 thoracic vertebrae (T1–T12), and 5 lumbar vertebrae (L1–L5), for a total of 25 channels per segmentation. To ensure spatial consistency and anatomical completeness, all segmentations are first resampled to an isotropic voxel spacing of $1\,\mathrm{mm}$. The center of the crop box is determined from the union (voxelwise sum) of all vertebral structures in each scan, and a fixed crop of $128^3$ voxels is applied for each patient.

For the knee dataset, we utilize the ShapeMedKnee dataset (Gatti et al., 2024) and extract 2000 3D segmentations of the left knee. The segmentations include the femur (Fe) and articular cartilage (Ca),

resulting in 3 channels per segmentation. To ensure spatial consistency and anatomical completeness, all segmentations are first resampled to an isotropic voxel spacing of $1\,\text{mm}$. A fixed crop size of $128^3$ voxels is applied for each patient.

## 8.3 LATENT DIFFUSION MODEL IMPLEMENTATION

For this study, we adapt the VAE and LDM architectures specified by Kadry et al. (2024). The VAE input and output channel counts are set to 11, corresponding to 10 distinct cardiac labels along with an additional channel for the background. The number of input channels for the LDM is set to 3 for unconditional sampling. The hyperparameters and training configuration for the VAE and LDM are listed in Table 2 and Table 3 respectively.

Table 2: Autoencoder hyperparameters

| Hyperparameter | Value |
|---|---|
| lr | $1 \times 10^{-5}$ |
| Epochs | 40 |
| Batch Size | 1 |
| Num. Channels | [16,32,64] |
| Num. Res. Blocks | 2 |
| Downscaling Factor | 4 |
| Recon. Loss Weight | 1 |
| KL Weight | $1 \times 10^{-6}$ |

Table 3: Diffusion model hyperparameters

| Hyperparameter | Value |
|---|---|
| **Training** | |
| lr | $2.5 \times 10^{-5}$ |
| Epochs | 50 |
| Batch Size | 1 |
| Num. Channels | [64, 128, 196] |
| Num. Res. Blocks | 2 |
| Num. Attn. Heads | 1 |
| Attn. Res. | 8, 4, 2 |
| $\sigma_{\text{data}}$ | 1 |
| $p(\sigma)$ mean | 1 |
| $p(\sigma)$ std | 1.2 |
| **Sampling** | |
| $\sigma_{min}$ | $1 \times 10^{-2}$ |
| $\sigma_{max}$ | 80 |
| $\rho$ | 3 |

## 8.4 GEOMETRIC GUIDANCE IMPLEMENTATION

### 8.4.1 GEOMETRIC MOMENT COMPUTATION

To ensure that the extracted components yield interpretable moments, we require the voxel grid values to be softly binarized, with one tissue channel approaching 1 while the others are close to 0. To achieve this, we apply a softmax function with a temperature of 1. During the computation of geometric moments, we observed that segmentations that are empty or nearly empty, particularly those with small components, lead to unstable gradients that significantly degrade the quality of generation. This instability arises because the centroid and covariance loss calculations utilize mass in the denominator. To mitigate this issue, we introduce a small amount of noise to the mass term whenever it appears in the denominator, thereby stabilizing the overall process. After computing all moments, we normalize the mass term by the total number of voxels $N = HWD$ such that the term represents volume fraction. Unless stated otherwise, we use 50 denoising steps.

### 8.4.2 GUIDANCE WEIGHT TUNING

We determine the weight factors $\lambda = [\lambda_0, \lambda_1, \lambda_2]$ for our geometric loss through tuning each loss in isolation. We tune for conditional fidelity while retaining reasonable generation quality metrics. The final weight values can be seen in Table 4.

Table 4: Geometric moment losses and their corresponding weight factors.

| Guidance Loss | Weight Factor $\lambda$ |
|:---:|:---:|
| $\mathcal{L}_{\text{size}}$ | $10^7$ |
| $\mathcal{L}_{\text{pos}}$ | $10^5$ |
| $\mathcal{L}_{\text{shape}}$ | $10^4$ |

### 8.5 BASELINE METHODS IMPLEMENTATION

- **Explicit Conditioning**: To ensure that the elements of $\mathcal{G}_{\text{exp}}$ are roughly between 0 and 1, we min-max normalize the masses $\mathcal{M}$, centroids $\mathcal{C}$, and normalized covariances $\mathcal{S}_n$ with values calculated from the real dataset (Table 5). The LDM input channel count is increased to accommodate the concatenated input. This method does not readily permit the use of dropout to train a diffusion model in an unconditional manner because the null condition is defined as zero—equivalent to the minimum moment values. We include explicit conditioning results for guidance weights smaller than 0 in Figure 4 for completeness.

- **Cross-Attention Conditioning**: Our initial tokens consist of 13-dimensional vectors representing the concatenation of mass $\mathcal{M}$, centroids $\mathcal{C}$, and normalized covariances $\mathcal{S}_n$. The tokens are then min-max normalized similar to explicit conditioning and embedded into a 256 dimensional vector for cross-attention. To embed the component index, we use a linear embedding layer. To embed the geometric moments, we use an MLP with three linear layers and apply a ReLU operation after the first and second layers. Both embeddings are added together and used to condition the U-Net with cross-attention, where we use 8 attention heads. To enable unconditional generation, we randomly drop each channel of $\mathcal{G}_{\text{cross}}$ with a probability of 0.1.

- **Implicit Conditioning**: To compute the ellipsoidal distance map, we use the centroids $\mathcal{C}$ and non-normalized covariances $\mathcal{S}$ for each component to compute the Mahalanobis distance (De Maesschalck et al., 2000) for each voxel position. We then apply a shifted sigmoid transform—with a slope of -0.5 and a bias of 1 to constrain the outputs between 0 and 1, and subsequently concatenate the resulting grid to the latents. To enable unconditional generation, we randomly drop each channel of $\mathcal{G}_{\text{imp}}$ with a probability of 0.1. One limitation of this approach is that the target mass can only be targeted indirectly through the non-normalized covariance term, which can be seen in the conditional fidelity plot for size in Figure 4.

Table 5: Normalizing constants for geometric moments during explicit and cross-attention based conditioning.

| Geometric Moment | Normalizing Minimum | Normalizing Maximum |
|:---:|:---:|:---:|
| $\mathcal{M}$ | $3.19 \times 10^{-3}$ | $1.3 \times 10^{-2}$ |
| $\mathcal{C}$ | $0$ | $1$ |
| $\mathcal{S}$ | $-1 \times 10^{-4}$ | $1 \times 10^{-2}$ |

## 8.6 ADDITIONAL EXPERIMENTAL DETAILS

- **Morphological evaluation metrics**: To compute the morphological metrics, the features are normalized by the mean and standard deviation of the real data. To calculate precision and recall, we use 5 neighbors.

- **Pointcloud evaluation metrics** To compute the point cloud metrics, we calculate MMD, COV, and NNA for every tissue label using 256 points sampled using farthest point sampling. The metrics are then averaged over the number of components. To compute the pointcloud distances, we approximate Earth Mover's Distance (EMD) through the Sinkhorn divergence (Feydy et al., 2019).

- **Disentangled Generation**: Disentangled generation is done by zeroing out the inactive loss weights. Exact configuration details are shown in Table 6. We use 50 denoising steps for all generated samples.

- **Compositional Generation**: Our compositional generation experiments vary the number of constrained substructures. The exact labels used for each experiment are detailed in Table 7. We use 100 denoising steps for all generated samples.

Table 6: Configuration details for the disentangled generation ablation study. Checkmarks ✓ indicate the associated weight factor $\lambda_i$ is active while × indicates the weighting factor is zeroed out.

| Guidance Loss | $\lambda_0$ | $\lambda_1$ | $\lambda_2$ |
|:---:|:---:|:---:|:---:|
| Uncond. | × | × | × |
| $\mathcal{L}_{\text{size}}$ | ✓ | × | × |
| $\mathcal{L}_{\text{pos}}$ | × | ✓ | × |
| $\mathcal{L}_{\text{shape}}$ | × | × | ✓ |
| $\mathcal{L}_{\text{geom}}$ | ✓ | ✓ | ✓ |

Table 7: Configuration details for the compositional generation study.

| Substructures | Labels |
|:---:|:---:|
| 0 | None |
| 1 | RV |
| 2 | RV, RA |
| 3 | RV, RA, PA |
| 6 | RV, RA, PA, LV, LA, Ao |

## 8.7 BIOMECHANICAL SIMULATION DETAILS

- **Biventricular Cropping**: As only myocardial tissue is available for the left ventricle, we approximate an RV myocardial wall by dilating the RV cavity mask to a constant thickness of 4 mm (2 voxels) corresponding to the clinical literature (Ho & Nihoyannopoulos, 2006). To crop the left and right ventricles at the base of the heart, we define a vector from the LV centroid to the LA centroid, and crop the ventricles by adjusting the position threshold along the defined direction.

- **Tetrahedral Meshing and Processing**: The segmentation is then converted into a surface mesh using marching cubes, with a voxel size of 2 mm. Tetrahedral mesh generation is performed using the open-source software `Gmsh` and `MeshLab`. The three anatomical models, large RV, baseline patient, and small RV (see Figure 11), are discretized into 39,780, 42,768, and 47,347 linear tetrahedral elements, respectively, with an average edge length of 2 mm.

- **Pressurization Simulation**: An in-house finite element method (FEM) solver, implemented in `Fortran` with `MPI`, is used for the simulations. The solver is based on the variational multiscale method, providing stabilized FEM formulations (Goraya et al., 2024; Kang et al., 2022). Simulation results are visualized using the open-source package `ParaView`.

The myocardium is modeled as a standard neo-Hookean material with a Young's modulus of 25 kPa and a Poisson's ratio of 0.4. Physiological pressure loads of 12 mmHg and 6 mmHg were applied to the LV and RV endocardium, respectively, corresponding to normal diastolic blood pressure. To constrain rigid body motion, zero-displacement Dirichlet boundary conditions are imposed at the base of the heart, while a stress-free Neumann boundary condition is applied on the pericardium.

The nonlinear finite-deformation elasticity problem is then solved using the Newton–Raphson (NR) method. A direct solver (MUMPS) is employed to solve the discretized algebraic system at each NR iteration, with a convergence tolerance set to $10^{-20}$ for the initial residual. Simulations are carried out on a cluster using 128 processors.

## 8.8 Dataset Scaling Analysis

We aim to understand the effect of dataset size on both generation quality and conditional fidelity under guidance. To this end, we train four additional autoencoder–diffusion model pairs at different split sizes, using 20%, 40%, 60%, and 80% of the original training set, while keeping the validation set fixed across all models. As shown in Figure 13, generation-quality metrics improve as the training dataset size increases up to 40%. In contrast, conditional fidelity under geometric guidance remains approximately invariant across dataset sizes.

## 8.9 Parametric Ellipsoid Dataset Analysis

To further characterize our geometric guidance procedure in isolation from complex anatomical variation, we construct a toy dataset of 3D two-channel ellipsoidal label maps with varying sizes, shapes, and positions. To generate each voxel map, we sample the ellipsoidal radii uniformly from $0$ to $0.5$, where $1$ corresponds to the full length of the voxel map. We additionally sample Euler angles uniformly from $0$ to $2\pi$, and choose the centroid to lie anywhere within the voxel map such that the ellipsoid is not cropped by the voxel boundaries. We generate $800$ training and $200$ validation label maps.

We then train a latent diffusion model with double the number of base channels to accommodate the large geometric variation in the dataset. We apply our geometric guidance method with centroid and covariance loss weights multiplied by a factor of 10. As shown in Figure 14, our geometric guidance framework can enforce precise geometric constraints on parametric ellipsoid geometries in a disentangled manner.

## 8.10 Autoencoder Reconstruction Fidelity Analysis

We aim to determine whether the conditional fidelity metrics and topological quality are lower-bounded by the VAE reconstruction error. We first auto-encode 24 seed label maps and measure conditional fidelity for size, position, and shape, as well as the Betti error for each anatomical structure. We then sample 24 label maps over 50 diffusion steps with and without right-ventricular geometric guidance. As summarized in Table 8, geometric guidance substantially improves conditional fidelity relative to unconditional sampling, while the resulting errors for position and shape remain above the VAE reconstruction error.

In terms of topology, we quantify quality using the Betti error, defined as the number of extra connected components relative to the expected topology (e.g., if the aorta is expected to be a single connected component but two are measured, the Betti error is 1). We observe that the VAE introduces only a small number of topological defects, whereas the unconditional diffusion model produces more frequent errors, especially for the aorta (Ao) and pulmonary artery (PA) labels. Finally, we find that geometric guidance can further increase the Betti error, particularly for the PA and inferior vena cava (IVC) labels.

## 8.11 Editing Scale Factor Analysis

We investigate how far the target right-ventricular mass can be scaled while still producing plausible samples. For each editing factor in $\{0.1, 0.5, 1.0, 2.0, 4.0\}$, we take 64 seed label maps, compute the RV mass, multiply it by the editing factor, and use the scaled mass as the conditioning target

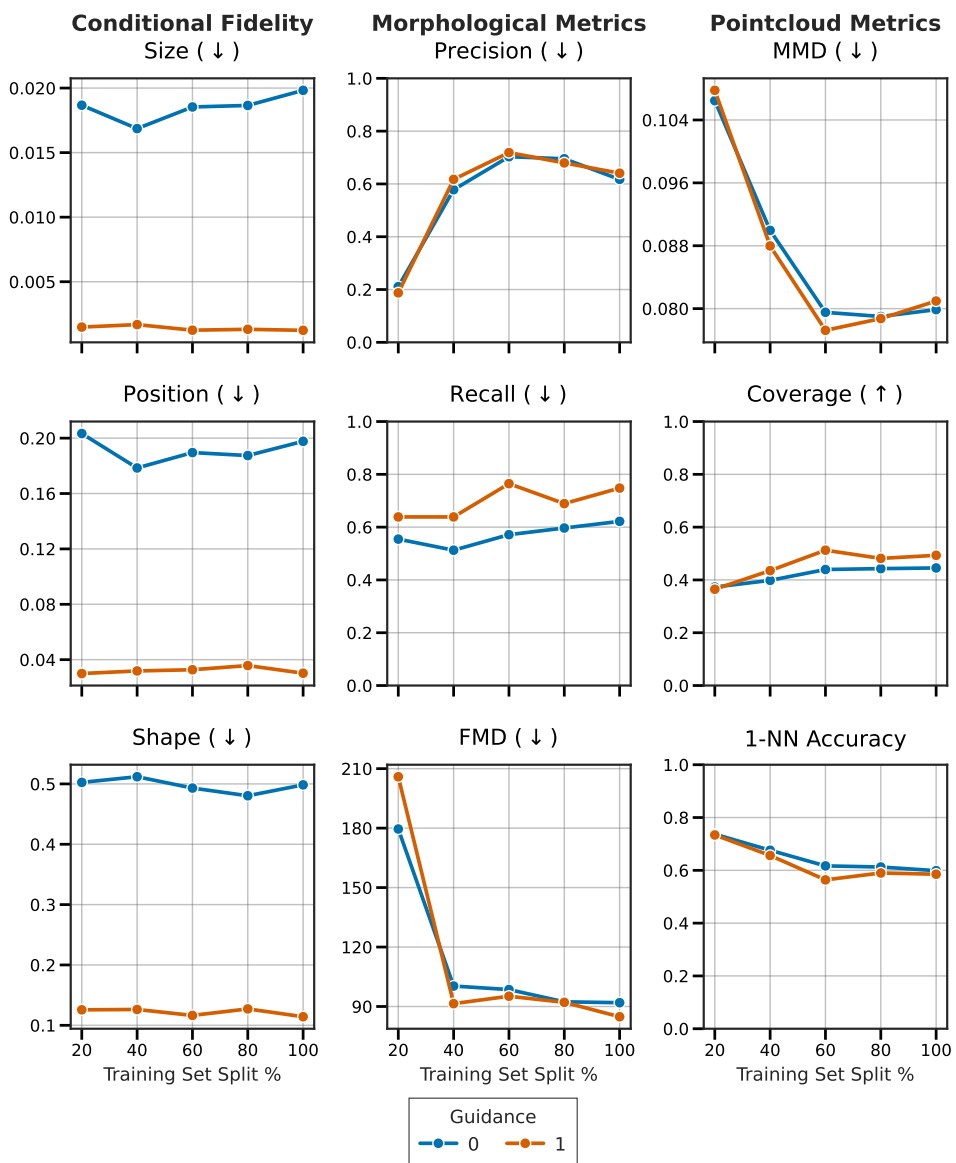

Figure 13: **Conditional fidelity is invariant to training set size, while generation quality metrics benefit from training set size up to a limit**. Line plots show conditional fidelity and generation quality for latent diffusion models trained on different-sized datasets. In this plot, the right ventricle is constrained.

for mass-only geometric guidance. As summarized in Table 9, decreasing the target mass (factors < 1) yields samples whose size error and distributional metrics remain close to the unedited case (factor = 1): size error increases moderately, and FMD and 1-NNA remain within the same order of magnitude as the baseline. In contrast, increasing the target mass beyond a factor of 2 leads to clear degradation: at a factor of 4, both the size error and FMD increase by more than one order of magnitude, and 1-NNA worsens, indicating that strong mass upscaling produces distorted label maps.

## 8.12 GEOMETRIC GUIDANCE WITH ALTERNATIVE MOMENT-FEATURES

In our main study, we demonstrated guidance by targeting the normalized second moment to control shape and orientation independently from size. We aim in this section to preliminarily demonstrate

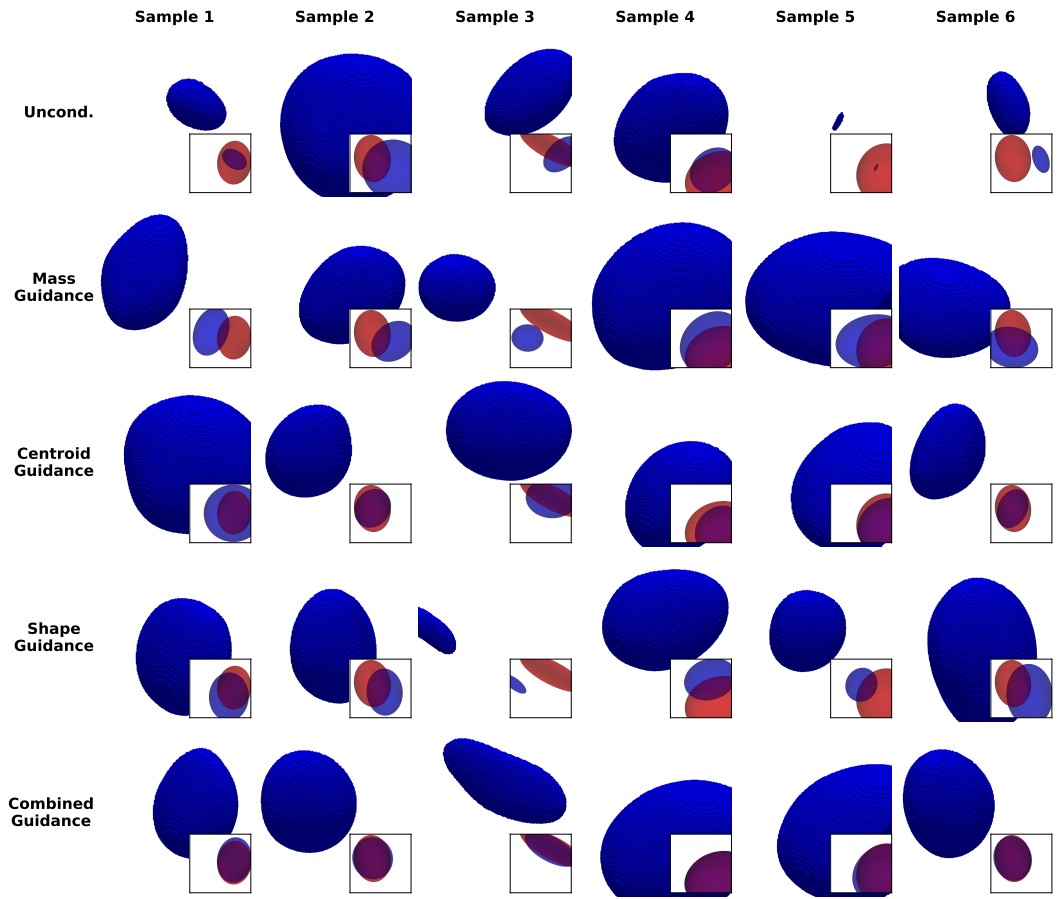

Figure 14: **Geometric guidance can control unconditional diffusion models of procedurally generated ellipsoids. We generate ellipsoidal label maps with varying loss-function combinations to achieve disentangled control.**

Table 8: Conditional fidelity and Betti error rates for reconstructed or synthetic label maps. Betti error is computed as the mean number of connected components minus 1. Values for size, position, and shape fidelity were multiplied by 1e5, 1e4, 1e4 respectively.

| Method | Cond. Metrics | | | Connected Component Betti Error | | | | | | | | |
| | Size | Pos. | Shape | Ao | PA | IVC | SVC | LA | RA | LV | RV | Myo |
|---|---|---|---|---|---|---|---|---|---|---|---|---|
| VAE Recon. | **4.74** | **2.54** | **21.53** | **0.125** | **0.0** | **0.125** | 0.042 | **0.042** | **0.083** | **0.0** | 0.083 | **0.0** |
| Unconditional | 106.03 | 168.75 | 512.42 | 1.208 | 0.583 | **0.125** | **0.0** | 0.083 | **0.083** | **0.0** | **0.0** | **0.0** |
| Guided | 9.34 | 25.61 | 81.77 | 1.458 | 1.042 | 0.417 | 0.042 | 0.083 | 0.333 | 0.042 | 0.083 | 0.042 |

that we can achieve fine-grained disentangled control of second-moment derived attributes such as extent, stretch, and orientation. We first decompose the covariance matrix as follows:

$$\mathcal{S} = v\mathbf{U}\Lambda^n\mathbf{U}^T, \tag{6}$$

where we define the extent $v \in \mathbb{R}$ as the trace of the eigenvalue matrix $\Lambda \in \mathbb{R}^{3\times3}$, and normalize $\Lambda$ by $v$ to obtain the anisotropic stretch $\Lambda^n = \Lambda/v$. Finally, orientation is represented by the eigenvectors $\mathbf{U} \in \mathbb{R}^{3\times3}$ derived from the decomposition.

We then define three new geometric losses, which consist of MSE losses for extent and stretch, as well as a dot product loss for orientation.

$$\mathcal{L}_{\text{extent}} = \mathcal{L}_{\text{MSE}}(v, \bar{v}), \quad \mathcal{L}_{\text{stretch}} = \mathcal{L}_{\text{MSE}}(\Lambda^n, \bar{\Lambda}^n), \quad \mathcal{L}_{\text{orient}} = \mathcal{L}_{\text{dot}}(\mathbf{U}, \bar{\mathbf{U}}). \tag{7}$$

Table 9: We generate label maps with mass-only geometric guidance applied to the right ventricle and artificially changing the target mass derived from the seed label map.

| | Cond. Metrics | Morph. Metrics | | | Pointcloud Metrics | | |
|---|---|---|---|---|---|---|---|
| **Editing Factor** | **Size** | **FMD** ($\downarrow$) | **Pr.** ($\uparrow$) | **Re.** ($\uparrow$) | **MMD** ($\downarrow$) | **COV** ($\uparrow$) | **1-NNA** |
| 0.1 | 84.06 | 126.0 | 0.14 | 0.34 | 13.51 | 0.295 | 0.826 |
| 0.5 | 19.95 | 59.6 | 0.55 | 0.50 | 10.51 | 0.406 | 0.713 |
| 1.0 | **12.60** | **40.3** | **0.59** | **0.73** | **9.26** | **0.520** | **0.567** |
| 2.0 | 18.55 | 167.5 | 0.16 | 0.84 | 11.08 | 0.430 | 0.745 |
| 4.0 | 732.10 | 1690.0 | 0.00 | 0.97 | 26.81 | 0.273 | 0.926 |

The dot product loss $\mathcal{L}_{\text{dot}}$ is computed as the mean misalignment between corresponding eigenvectors from $\mathbf{U}$ and $\bar{\mathbf{U}}$,

$$\mathcal{L}_{\text{dot}}(\mathbf{U}, \bar{\mathbf{U}}) = \frac{1}{3} \sum_{i=1}^{3} \left(1 - |\mathbf{u}_i^\top \bar{\mathbf{u}}_i|^2\right), \tag{8}$$

where $\mathbf{u}_i$ and $\bar{\mathbf{u}}_i$ denote the $i$-th columns of $\mathbf{U}$ and $\bar{\mathbf{U}}$, respectively, and the absolute value enforces sign-invariance of eigenvector alignment.

With these losses, we conduct a disentangled generation experiment where we sample 32 label maps for each loss ablation setting, with the loss weightings detailed in Table 10. Conditional fidelity for extent and stretch is quantified using the mean absolute error, while conditional fidelity for orientation is quantified using the dot-product loss directly. As shown in Table 11, geometric guidance based on second-order derived features can be applied in a disentangled manner. For example, orientation-only guidance achieves a smaller orientation error while maintaining extent and stretch fidelity comparable to unconditional sampling.

Table 10: Second order moment losses and their corresponding weight factors.

| **Guidance Loss** | **Weight Factor** $\lambda$ |
|---|---|
| $\mathcal{L}_{\text{extent}}$ | $10^5$ |
| $\mathcal{L}_{\text{stretch}}$ | $10^4$ |
| $\mathcal{L}_{\text{orient}}$ | $10^2$ |

Table 11: We enable disentangled control over geometric features derived from decomposing the second moment into extent, stretch, and orientation. Conditional fidelity metrics for extent and stretch, as well as MMD values were multiplied by 1e3.

| | Cond. Metrics | | | Morph. Metrics | | | Pointcloud Metrics | | |
|---|---|---|---|---|---|---|---|---|---|
| **Method** | **Extent** | **Stretch** | **Orient.** | **FMD** ($\downarrow$) | **Pr.** ($\uparrow$) | **Re.** ($\uparrow$) | **MMD** ($\downarrow$) | **COV** ($\uparrow$) | **1-NNA** |
| None | 2.14 | 42.30 | 0.21 | 67.63 | 0.44 | **0.80** | 10.22 | 0.484 | 0.563 |
| Extent Only | **1.42** | 41.85 | 0.20 | 61.99 | 0.50 | 0.78 | 10.12 | 0.459 | **0.566** |
| Stretch Only | 2.26 | **3.60** | 0.23 | **60.53** | 0.53 | **0.80** | 10.08 | 0.491 | 0.559 |
| Orient Only | 1.94 | 41.29 | **0.0064** | 60.81 | **0.59** | 0.77 | **10.08** | **0.525** | 0.564 |

## 8.13 MORPHOLOGICAL ANALYSIS

We represent size as the mass of each substructure. Position is represented by the centroid x-coordinate. To characterize shape, we extract the largest eigenvalue and its associated eigenvector from the covariance matrix. Orientation is represented by the polar angle of the principal axis (in spherical coordinates), while elongation is defined as the ratio between the largest eigenvalue and the second-largest eigenvalue.

Additional morphological plots are presented below. In Figure 16, we show that geometric guidance better aligns the distribution of geometric features when comparing real and synthetic anatomies.

Figure 15 shows that all geometric-control methods can recapitulate the morphological distribution exhibited by the real data.

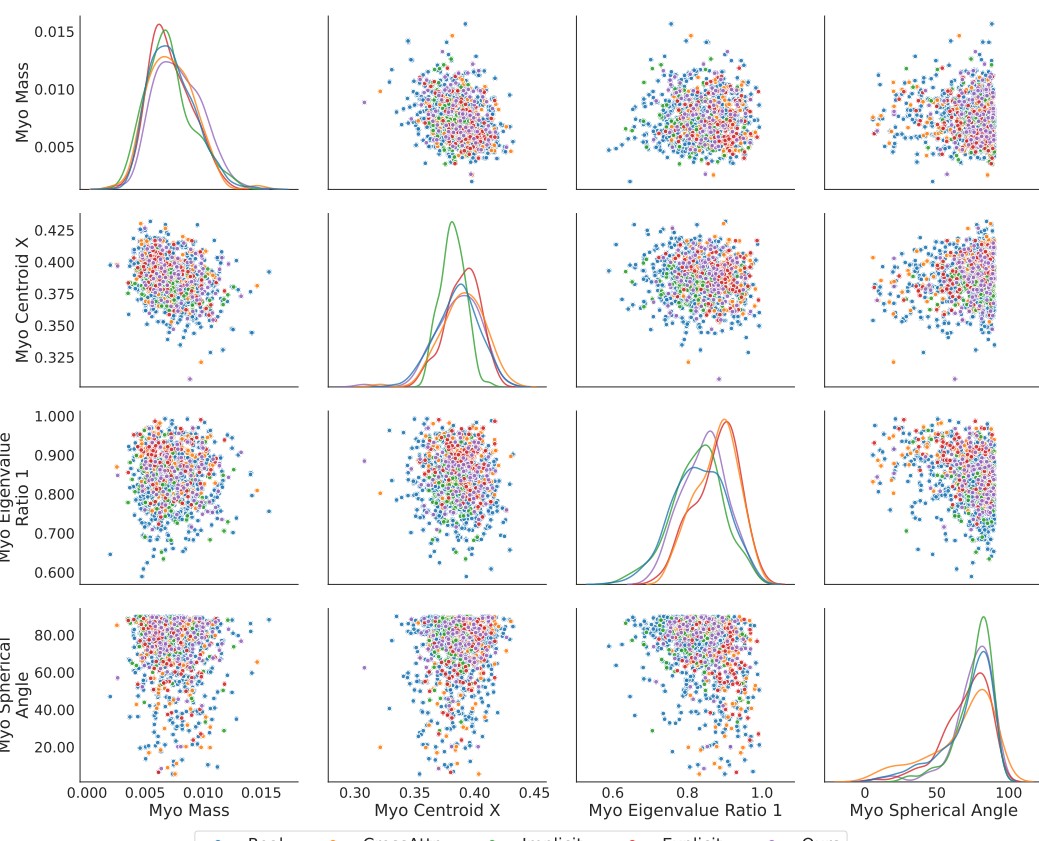

Figure 15: **Geometric guidance can help recapitulate morphological distributions.** Pair plot shows kernel density estimate plots (diagonals) and pairwise scatterplots (off-diagonals) for various morphological metrics. We plot metrics for anatomies generated through conditional baselines and geometric guidance (ours). In this plot, the myocardium is being constrained.

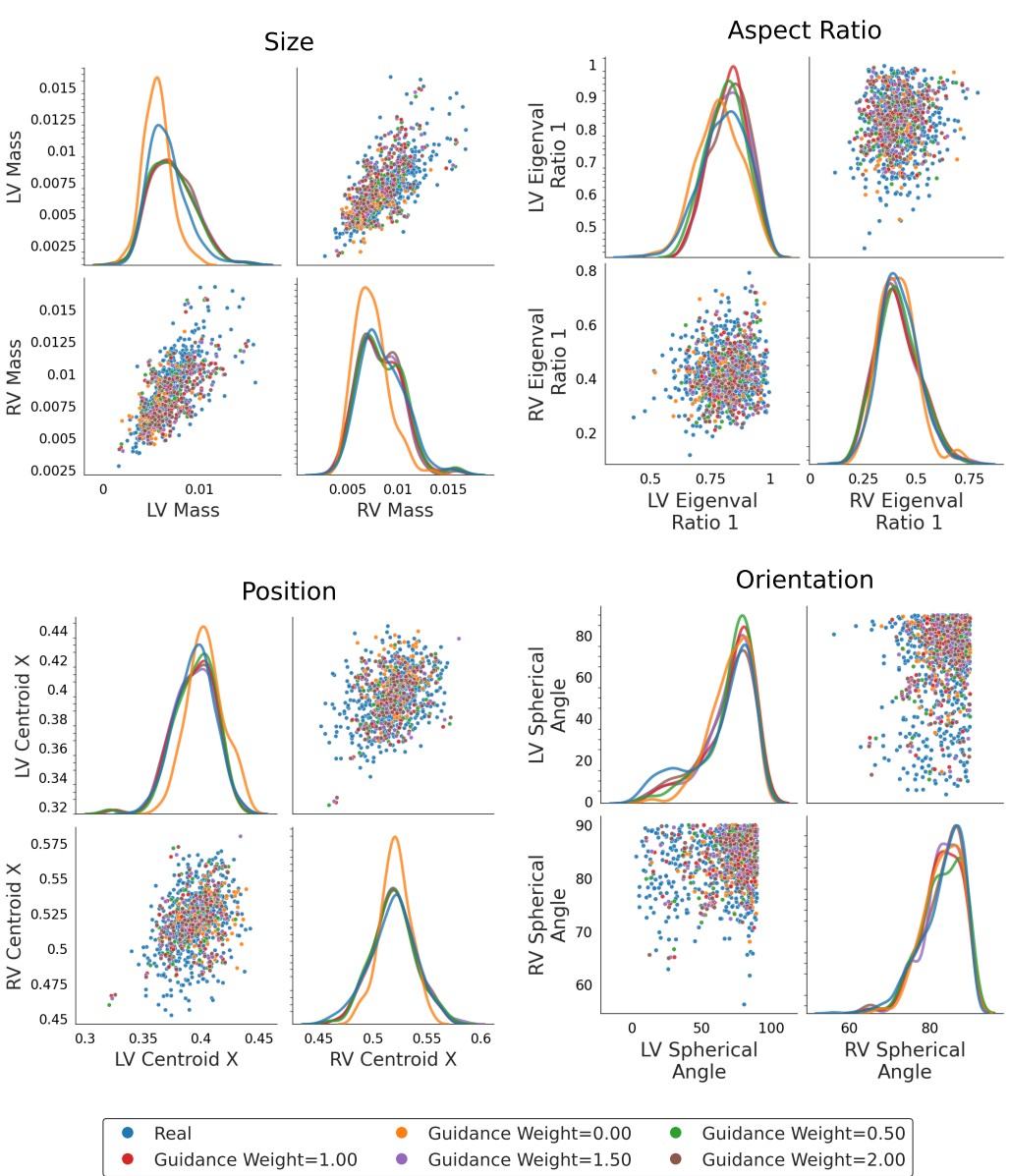

Figure 16: **Geometric guidance improves morphological distribution similarity between real and synthetic anatomy.** Pair plot shows morphological relationships for mass (top left panel), centroid (bottom left panel), normalized axis lengths (top right panel), and orientation (bottom right panel), where the myocardium labels are being constrained. Diagonal plots show kernel density estimates (LV vs RV), off-diagonal plots show pairwise scatterplots.

