# OpenReview forum: "CardioComposer: Leveraging Differentiable Geometry for Compositional Control of Anatomical Diffusion Models"
_ICLR.cc/2026/Conference — ICLR 2026 Poster_

### Official Review · Reviewer_gDx6 · 2025-10-31

**Soundness:** 4
**Presentation:** 4
**Contribution:** 3
**Rating:** 8
**Confidence:** 3

**Summary:**

This paper presents a novel framework for imposing geometric constraints models without model retraining. This is achieved by introducing a differentiable geometric loss during inference. The loss aims to match the geometric moments (size, position, and shape) between the predictions and targets. The framework was evaluated on the Totalsegmentator dataset and compared to several baseline methods that integrate the geometric information either explicitly or implicitly.

**Strengths:**

1. The paper presents a novel framework for anatomical structure generation that enables flexible control over geometric attributes during inference without requiring retraining, a notable advantage over conventional conditional generative approaches.
2. The proposed geometric loss formulation is both interpretable and modular, leveraging moment-based constraints to disentangle control over size, shape, and position. The method further supports compositional multi-part control and demonstrates utility in simulation-based downstream tasks.
3. The authors have shown model's robustness against hyperparameters and non-convex shapes.

**Weaknesses:**

The method introduces a novel formulation in Eq. (4) to define and normalize geometric moments, offering a clear way to separate size, shape, and orientation characteristics. However, the overall idea of combining a Latent Diffusion Model with test-time constraints has been explored in prior works, so the main novelty lies in the specific geometric moment formulation rather than the general framework.

**Questions:**

Good paper with clear practical relevance and a well-motivated clinical application. While it does not introduce major technical innovations, it effectively addresses a real-world problem and demonstrates solid empirical value in the medical field.

---

> ### Author Response · Authors · 2025-11-22
> **Response by Authors**
>
> Thank you for the positive review.
> > "the overall idea of combining a Latent Diffusion Model with test-time constraints has been explored in prior works, so the main novelty lies in the specific geometric moment formulation rather than the general framework."
>
> We agree that test-time guidance for latent diffusion models has been extensively explored in various domains, including text-to-image generation, inverse problems (e.g., deblurring and denoising), robotics planning, and protein generation.
>
> We also view our main contribution is integrating differentiable geometric moments into the guidance signal, which induces gradient structures that differ from, for example, neural classifier–based guidance. In addition, by doing this, we unlock a simple, modular design space that separates size, position, and shape, and we show that these controllable factors are directly aligned with clinically relevant requirements for cardiovascular device design.

---

### Official Review · Reviewer_WPU8 · 2025-10-31

**Soundness:** 2
**Presentation:** 3
**Contribution:** 3
**Rating:** 4
**Confidence:** 4

**Summary:**

The paper proposes CardioComposer, a 3D method enabling differentiable, geometry-aware control of anatomical diffusion models. The main contribution is to introduce geometric moment-based losses that are differentiable, which can guide sampling to produce anatomically realistic and geometrically controllable multi-class label maps of the heart.

**Strengths:**

1. Using geometric moments as differentiable constraints for guiding diffusion models of anatomy is novel.
2. The paper presents clear mathematical formulation of 0th, 1st and 2nd order geometric moments.
3. The paper includes baselines such as explicit and implicit concatenation, cross-attention.
4. The paper evaluates different combinations of proposed loss as the ablation study.

**Weaknesses:**

1. The experiment is focusing on cardiac structures with limited samples (less than 600 in total). The generalizability and scalability remain concerns.
2. Are the unconditional latent diffusion models (3D VAE and 3D UNet) trained from scratch?
3. In the Figure 7, for unconditional generation, why does the implicit fail but the proposed one seems to be able to generate a much more realistic map? Further explanation or more visualization would strengthen this observation.

**Questions:**

Will the code be released to the public?

---

> ### Author Response · Authors · 2025-11-22
> **Response by Authors**
>
> Thank you for the review, addressing these comments has been very fruitful through inspiring followup experiments and analyses that have improved the manuscript. Our updated manuscript and appendix can be downloaded above.
>
> >  "The experiment is focusing on cardiac structures with limited samples (less than 600 in total). The generalizability and scalability remain concerns."
>
> We understand this comment to reasonably ask about 1) the generality of our framework outside of cardiac data and 2) how dataset size impacts the performance.
> - To demonstrate **generalizability**: We present **Figure 12**, where we curate three new datasets of anatomical segmentations exhibiting a wide variety of forms and structures, and apply our geometric guidance framework with the same hyperparameters and loss weightings. Specifically, we retrain all our models on datasets relating to the branched ascending aorta (500 patients), spinal column (800 patients), and the femur (2000 patients). Additionally, we analyze a fifth toy dataset as suggested by reviewer `aon8` in **Figure 14**, where we train a latent diffusion model on procedurally generated ellipsoids and apply geometric guidance to achieve disentangled control.
> - To analyze **scalability**: We present a data scaling analysis in **Figure 13**, where we train separate latent diffusion models on increasing train set sizes while using the exact same validation set and architecture. Interestingly, we found that generation quality plateaus at 40% train set size (around 190 patients). We also found conditional fidelity to be invariant to train set size, exhibiting similar conditional fidelity metrics at low and high train set sizes.
>
> > "Are the unconditional latent diffusion models (3D VAE and 3D UNet) trained from scratch?"
>
> We confirm that all VAEs and LDMs are trained from scratch, despite being trained on ~500-2000 patients. In contrast to more conventional generative modeling of natural images, which typically require much larger train sets, we believe training a generative model of anatomical segmentations is much easier for the following reasons:
> - Lack of low-level texture information: the main variability in anatomical multi-class segmentation datasets is not voxel-level textures but rather higher-level features like shape and position. This makes VAE-compression straightforward, where we achieve 95% dice scores with little effort.
> - Data variability: most anatomical datasets exhibit variability that is important but subtle. For instance, the left ventricle in the cardiac dataset is most often located in the center of the voxel grid and is often adjacent to a right ventricle. This allows the latent diffusion model to focus only on the variation between patients. As such, we fully expect a 'foundation model' for anatomic generation that models a variety of anatomical systems to require significantly more data due to higher variability.
>
> > "In the Figure 7, for unconditional generation, why does the implicit fail but the proposed one seems to be able to generate a much more realistic map? Further explanation or more visualization would strengthen this observation."
>
> Thank you for this interesting question. We believe the decrease in performance for the implicit model is due to conditional dropout. More specifically, we trained the implicit model by conditioning on 6 ellipsoidal heatmaps corresponding to each substructure, where each heatmap was independently dropped out with some probability. This means that the most common conditioning signal seen was the one with no dropout (all 6 channels were seen), and the least common signal was one in which all the structures were dropped out (all empty heatmaps), which exactly corresponds to the good and bad generation quality, respectively. We also add the following text in Section 5.6 to explain further:
> - "Because the implicit conditional baseline is trained with independent dropout over six ellipsoidal conditioning channels, the fully conditioned case (all channels present) is vastly more frequent than the unconditional case (all channels empty). As a result, unconditional sampling corresponds to the rarest training configuration and yields degraded anatomical quality in Figure 7 and Table 1."
>
> > "Will the code be released to the public?"
>
> We are happy to announce that we will release the code for CardioComposer upon publication. We will specifically include a separate module and tutorial for our differentiable geometric measurement functions for other applications outside cardiac generation.

---

### Official Review · Reviewer_aon8 · 2025-11-01

**Soundness:** 4
**Presentation:** 4
**Contribution:** 3
**Rating:** 8
**Confidence:** 4

**Summary:**

This paper introduces "CardioComposer," a novel inference-time guidance framework designed to control unconditional diffusion models for the generation of 3D multi-class anatomical segmentations. The primary problem it addresses is the trade-off between geometric controllability and realism in generative models for cardiovascular anatomy, which is a critical component for clinical research and in silico medical device evaluation.

The core contribution is the development of differentiable measurement functions based on voxel-wise geometric moments. These functions compute interpretable geometric attributes—specifically size (from zeroth-order moments), position (from first-order moments), and shape (from scale-normalized second-order moments)—from the generated label maps. The gradients from a loss function comparing these measured moments to target moments are then used to guide the diffusion sampling process at inference time.

**Strengths:**

1) The core contribution—a flexible, inference-time guidance mechanism based on differentiable geometric moments for 3D multi-class segmentations—is somewhat novel. This method directly tackles a significant bottleneck in the use of generative models for scientific simulations, where interpretable control over specific geometric properties is paramount. The ability to generate "digital siblings" for counterfactual simulations is interesting.

2) The paper is well-structured and clearly written. Figures are informative (outstanding), though some are dense. The related work section is comprehensive. The writing is precise.

3) Extensive evaluation: 200 samples per method, 5+ baselines, 3 metric families (morphological, pointcloud, conditional fidelity). Strong ablations: disentanglement, compositional control, guidance weight tuning. Real-world validation: Used to edit patient anatomy for biophysical simulation (RV volume vs. wall displacement), showing causal form-function modeling.

4) The method is technically sound, with a clear formulation of the geometric moment losses (size, position, and scale-normalized shape). The experimental validation is rigorous.

**Weaknesses:**

The authors are commendable for their transparent limitations section. Some weaknesses:

1) The framework relies on weights $\lambda_i$ for the aggregate loss and a guidance weight $w$. The paper notes $\lambda_i$ require "experimental tuning" and provides the final values in the appendix37. However, the sensitivity to these weights is not explored. How robust is the method to changes in these hyperparameters?

2) The authors rightly admit that the underlying diffusion model can generate topologically incorrect anatomies, which is a critical failure for biophysical simulations. The proposed solution is post-hoc filtering. It is unclear if the proposed geometric guidance affects the rate of these topological errors

3) While framed as general, all experiments are on cardiac data.

4) In Figure 4, while the proposed method  generally maintains high-quality metrics, there appears to be a slight degradation in realism at high guidance weights ($w > 1.5$)

**Questions:**

1) How sensitive is the method to VAE reconstruction error?

2) Can you control orientation independently of shape?

3) What happens under extreme geometric targets (e.g., 10× mass)?

---

> ### Author Response · Authors · 2025-11-22
> **Response by Authors**
>
> Thank you for the review. Addressing these comments has been very fruitful through inspiring follow-up experiments and analyses that have improved the manuscript. Our updated manuscript and appendix can be downloaded above.
> > "However, the sensitivity to these weights is not explored. How robust is the method to changes in these hyperparameters?"
>
> We apologize for the confusion. The requested sensitivity analysis for the loss weightings (lambda_i notation) reported in the appendix is Figure 9 in the original manuscript. We created the plot under different labels within the legend (Mass only, Centroid only, etc.). We found that our loss weightings influence conditional fidelity and generation quality independently, enabling easy tuning.
>
> > "The underlying diffusion model can generate topologically incorrect anatomies, which is a critical failure for biophysical simulations. The proposed solution is post-hoc filtering. It is unclear if the proposed geometric guidance affects the rate of these topological errors"
>
> We add **Table 8**, where we conduct an analysis to isolate the influence of VAE reconstruction, unconditional sampling, and geometric guidance on topological quality. Interestingly, we show that VAE reconstruction minimally introduces topological defects. On the other hand, both unconditional sampling and geometric guidance can increase the rates of topological defects for some labels, while exerting negligible effects on other labels. We discuss this in the newly created **Section 8.10**.
>
> > "While framed as general, all experiments are on cardiac data."
>
> Thank you for this suggestion. To demonstrate generality outside of cardiac data, we construct three entirely new datasets of anatomical segmentations and apply our geometric guidance framework with the same hyperparameters. We retrain all our models on datasets relating to the branched ascending aorta, spinal column, and the femur. The results are now visualized in **Figure 12.**
>
> > "In Figure 4, while the proposed method generally maintains high-quality metrics, there appears to be a slight degradation in realism at high guidance weights
>
> We agree that, as with most guidance frameworks, improperly tuning the guidance weight can degrade generation quality metrics. We do, however, believe that experimental tuning is fairly painless due to the following reasons:
> - Loss weightings do not significantly interact (Figure 9), which means that tuning can be done for each weight in isolation.
> - Loss weightings seem to be robust when transferred to different anatomical datasets, where we have shown in Figure 12 that the same loss weightings can be applied out of the box. In the case of our procedurally generated toy dataset of ellipsoids, we only had to increase the position and shape weights by 10x to improve the results.
>
> > "How sensitive is the method to VAE reconstruction error?"
>
> Thank you for correctly pointing out that VAE reconstruction is also imperfect and may set a 'lower limit' for conditional fidelity. We therefore add **Table 8**, where we compare the conditional fidelity obtained from just reconstructing the seed label maps with the VAE, against sampling with and without geometric guidance. We find that VAE reconstruction error contributes to, but does not dominate the conditional fidelity error of the guided generations. We discuss this in the newly written **Section 8.10**.
>
> > "Can you control orientation independently of shape?"
>
> We provide **Table 12** to show that orientation control is possible by applying a dot product-based loss on the decomposed eigenvectors. We explain this in **Section 8.12**, in which we decompose the covariance matrix into the following:
> - Extent, which is defined as the trace of the covariance matrix.
> - Stretch, which is a vector of normalized eigenvalues indicating the aspect ratio of the substructure.
> - Orientation, which are the eigenvectors indicating the relative orientation of the substructure.
>
> As mentioned, we show that using these features as targets enables disentangled generation, where, for example, we synthesize label maps that have lower orientation loss but high levels of extent and stretch loss. We do, however, note that in the case of orientation control, the level of control depends on the natural anisotropy of the targeted substructure. A perfectly symmetric sphere does not have a canonical orientation when eigendecomposed, as many different eigenvectors describe the covariance equally well.
>
> > "What happens under extreme geometric targets (e.g., 10× mass)?"
>
> Thank you for this interesting suggestion. We conduct a scale-factor sweep for multiple geometric targets and report the results in **Table 9**. We find that reducing the target mass with a scale factor smaller than 1 produces reasonable label maps up to 10x shrinking, but increasing the target mass begins to produce distorted label maps at 4x expansion. We discuss this in the newly written **Section 8.11**.

---

### Official Review · Reviewer_fX4P · 2025-11-02

**Soundness:** 3
**Presentation:** 4
**Contribution:** 3
**Rating:** 6
**Confidence:** 4

**Summary:**

CardioComposer adds a geometric control mechanism to an unconditional diffusion model at inference time, using differentiable moments—volume (0th), centroid (1st), and scale-normalized covariance (2nd)—to control the size, position, and shape of cardiac parts.
No retraining is required: each denoising step decodes a segmentation, measures geometry, computes a loss to target values, and backpropagates its gradient to adjust the latent, enabling disentangled and composable control.
Trained on 596 multi-label cardiac segmentations, the method achieves stable single- and multi-part control, geometric inpainting, and biomechanical simulation compatibility.

**Strengths:**

1. **Clinical Significance**
The paper addresses a clinically relevant issue—how to create anatomically realistic yet controllable 3D cardiac models.

2. **Flexible Post-training Optimization**
The proposed inference-time optimization framework is elegant and practical. By introducing differentiable geometric moments as guidance losses, the method allows users to customize geometric targets on demand without retraining or modifying network weights. This design provides an interpretable and lightweight mechanism for controllable shape generation.

3. **Comprehensive Experimental Validation**
The experiments are thorough and convincing. They demonstrate geometric controllability across multiple substructures, evaluate both single- and multi-part guidance, and further validate the realism of generated shapes using distributional metrics and biomechanical simulations. Together, these results strongly support the effectiveness and robustness of the proposed approach.

4. **Clarity and Completeness of Presentation**
The paper is clearly written and technically detailed.

**Weaknesses:**

1. **Lack of comparison with existing conditional shape generation methods.**

While the paper clearly articulates the benefits of inference-time guidance, it does not compare against established conditional generative models trained for geometric control (e.g., de Wilde et al., 2025 (https://arxiv.org/abs/2504.03313); Kadry et al., 2025 (https://www.nature.com/articles/s41746-024-01332-0), A-SDF (https://arxiv.org/abs/2104.07645). Including such baselines would better quantify the extent to which performance or controllability is improved by the proposed guidance relative to conditional training.

2. **Absence of simulated validation data.**

Although the authors compute geometric attributes (mass, centroid, covariance) from real cardiac segmentations, the evaluation relies solely on real anatomies. A complementary experiment on simulated or procedurally generated geometries—with known ground-truth attributes—would allow denser and more diverse target sampling, providing clearer evidence of control accuracy, disentanglement, and robustness.

3. **Limited explanation for Figures 6 and 8.**

The pair-plot visualizations are intriguing but under-explained: it remains unclear which visual trends (e.g., sharper peaks, denser clusters) indicate successful guidance.


4. **Code is not available**

**Questions:**

1. The generation quality appears to depend on the amount of training data available. How large a dataset is required for the diffusion model to achieve stable and realistic shape synthesis?

2. Apart from the 0th- and 1st-order moments, are there other geometric features that could serve as control signals?

---

> ### Author Response · Authors · 2025-11-22
> **Response by Authors**
>
> Thank you for the review. Addressing these comments has been very fruitful, inspiring follow-up experiments and analyses that have improved the manuscript. Our updated manuscript and appendix can be downloaded above.
>
> > "While the paper clearly articulates the benefits of inference-time guidance, it does not compare against established conditional generative models trained for geometric control"
>
> We understand this as a request to compare our inference-time framework with appropriate conditional training baselines. If so, then we apologize for the miscommunication, as we already implement the explicit, implicit, and cross-attention baselines in the paper as the conditional diffusion training baselines, enabling us to directly quantify our method against conditional training for the same diffusion architecture.
>
> We further see many of the baselines suggested as special cases of our implemented baselines:
> - For example, we view the approach of de Wilde et al. as using explicit conditioning applied to auto-decoder architectures, which we show in our study to be competitive in controlling size but not position or shape.
> - The other auto-decoder-based approach by Mu et al. also uses explicit conditioning to disentangle category-based generation from articulation by concatenating shape codes and articulation codes to generate single objects. These approaches are difficult to implement for multi-class cardiac generation, where all tissues are present in every sample, rather than between samples.
> - Finally, we believe that the approach of Kadry et al., while formulated for diffusion models, is mainly concerned with controlling unconditional diffusion models for partial diffusion (what they call perturbational editing) and inpainting (what they call localized editing). These tasks are different from controlled synthesis targeting specific geometric attributes.
>
> > "A complementary experiment on simulated or procedurally generated geometries—with known ground-truth attributes—would allow denser and more diverse target sampling, providing clearer evidence of control accuracy, disentanglement, and robustness."
>
> Thank you for this insightful suggestion. We agree that our method could be relying on the unconditional diffusion model using the natural correlations within our datasets to artificially improve our results.
>
> As requested, we provide **Figure 14** and **Section 8.9**, in which we have created a dataset of procedurally generated ellipsoid segmentations with no correlations between size, position, and shape. This allows us to properly isolate the influence of our method from anatomical variability. We then train a separate latent diffusion model for this toy dataset and demonstrate the capacity of our framework for disentangled geometric control.
>
> > "The pair-plot visualizations are intriguing but under-explained: it remains unclear which visual trends (e.g., sharper peaks, denser clusters) indicate successful guidance."
>
> We enhance Section 5.5 to explain disentangled generation in Figure 6 with the following text:
> - "For example, mass-only guidance produces a narrow peak in the mass marginal while the other morphology metrics remain broad, whereas applying all geometric losses collapses all marginals to narrow peaks at their target values."
>
> We enhance Section 5.6 to explain compositional generation in Figure 8 with the following text:
> - "For instance, when guidance is applied to a single substructure, the pair plots show a sharp concentration around the target value for the right ventricle, while the remaining structures retain broad distributions. When three substructures are guided simultaneously, the corresponding morphological marginals all collapse to narrow peaks at their target geometric values. Finally, Figure 8 shows that our guidance framework applies to complex, non–star-shaped geometries, including curved and branching substructures, as well as Boolean unions involving multiple tissue classes considered as a single substructure (e.g., both vena cavas or all chambers)."
>
> >  "Code is not available"
>
> We intend to release the code for CardioComposer upon publication. We will specifically include a separate module and tutorial for our differentiable geometric measurement functions for other applications outside cardiac generation.

---

### Author Response · Authors · 2025-11-22
**Overall Response by Authors**

We thank all reviewers for their help in significantly improving the quality of our manuscript. We are delighted that our paper contained comprehensive evaluations and ablations (`fX4P`, `aon8`, `WPU8`, `gDx6`), was clearly written (`fX4P`, `aon8`), and well formulated (`aon8`, `WPU8`). We also appreciated that our methodology was recognized in terms of its novelty (`aon8`, `WPU8`, `gDx6`), flexibility (`aon8`, `WPU8`, `gDx6`), and clinical significance (`fX4P`, `aon8`, `gDx6`).

In addition to the individual responses, we updated the manuscript that can be downloaded above with **new figures and results**, as well as writing improvements. All revised text in the manuscript appears in blue. New figures and tables have captions colored in blue. We summarize the main experiments and analyses resulting from reviewer suggestions below:

- **Expanded Set of Anatomical Datasets** (`aon8`, `WPU8`): To demonstrate the generality of our method outside of cardiac data, we construct three more patient-specific anatomical datasets corresponding to vascular, spinal, and femoral systems. We train separate models for each dataset and demonstrate that geometric guidance transfers out-of-the-box to these datasets in **Figure 12**.
- **Procedurally Generated Geometry Analysis** (`fX4P`): To disentangle our control framework from complex anatomical variation, we construct a fifth toy dataset of simple ellipsoidal segmentations. We demonstrate disentangled guidance with our framework in **Figure 14**.
- **Dataset Scaling Analysis** (`fX4P`, `WPU8`): To understand the impact of training data size, we conduct a dataset scaling analysis whereby we train on different training set sizes. We show in **Figure 13** that unconditional generation quality stops improving after using 40% of our training set, while conditional fidelity is invariant to training set size.
- **VAE Reconstruction Analysis** (`aon8`): To understand the influence of VAE reconstruction, we measure conditional fidelity after VAE auto-encoding, unconditional sampling, and guided sampling. We show that VAE reconstruction contributes minimally to conditional fidelity in **Table 8**.
- **Topological Fidelity Analysis** (`aon8`): To quantify the influence of geometric guidance on topological fidelity, we measure the connected component error after VAE auto-encoding, unconditional sampling, and guided sampling. We visualize these results in **Table 8**.
- **Geometric Decomposition Analysis** (`fX4P`, `aon8`): To demonstrate the ability of geometric guidance to constrain more subtle geometric features such as stretch and orientation, we further decompose the second moment into extent, stretch, and orientation, using these features as targets to guide label map generation. We preliminarily show that our framework achieves disentangled generation in **Table 11**.
- **Target Feature Scaling Analysis** (`aon8`): To gauge how well we can manually edit target geometric features, we sweep over various scaling factors for our target size during generation. We show in **Table 9** that our generations remain plausible until our target mass is 4 times that of the seed label map.

All other points are addressed in the individual responses below. We greatly appreciate the feedback and welcome further suggestions to improve the manuscript.

---

### Meta-Review · Area_Chair_giq9 · 2026-01-05

**Summary:**

This paper provides a test-time approach to sampling anatomical models with differentiable geometric constrains.

- Reviewers (fX4P, aon8, gDx6) agree that constraining diffusion models during inference time by encoding the geometry of shape guidelines (position, shape, size) of ellipsoids is an elegant approach that could be useful in a variety of important settings, such as in the context of digital twins (aon8).
- Reviewers agree that, while constraining diffusion models with some geometric prompts at inference time is not entirely novel (see gDx6), there is novelty in the specific way such geometry is encoded in the presented approach (see aon8 and gDx6)
- Presentation is clear and compelling (see fX4P, aon8, and gDx6).

**Reviewer Concerns:**

- Limited novelty: rev. gDx6 pointed to the fact that test-time constrained generation via diffusion models is not novel. This is acknowledged by the authors.
- Comparisons with baselines: rev fX4P requested comparison to other shape-constrained methods. This is partially addressed by the authors' rebuttals: they had already included methods similar to the ones requested (e.g. that of Wilde et al), or not comparable given that they are tricky to implement or relevant in other tasks.
- Revs aon8 and fX4P requested ablation studies and further empirical evidence, which was provided during the rebuttal (via a toy ellipsoids, dataset scaling, topological fidelity and orientation control).
- A concern is raised on the topological correctness of the produced reconstructions (see rev aon8). in the added experiments, the authors show that indeed some mistakes are introduced with their guidance/constrained generation. These error rates are reported and discussed.
- Lack of code is raised by two reviewers (fX4P and WPU8). Authors argue it will be released upon acceptance.

**Reviewer Scores:**

- Rev fX4P was borderline positive and confident (4). Their key concerns (lack of comparisons, lack of ground-truth, lack of code, clarifications of figures) were mostly addressed. In light of this, I anticipate they would have maintained or increased their score.
- Rev aon8 was clearly positive and confident. The added experiments, sensitivity analysis, reporting topological errors, would have contributed to strengthening their support.
- Rev WPU8 was borderline negative, but also mentioned that "would not mind if accepted". Most comments from this reviewer were clarifications in nature, rather than core concerns. They would have maintained or increased their score.
- Rev gDx6 was also positive and moderately confident; main critique being the limited novelty. This is not contented by the authors; likely maintaining the score.

All in all, reviewers were generally positive, with two strongly positive reviewers. I am thus recommending acceptance.

---

### Decision · Program_Chairs · 2026-01-26

Accept (Poster)